# Fairness-Aware Multi-view Evidential Learning with Adaptive Prior

**Haishun Chen**[1], **Cai Xu**[1], **Jinlong Yu**[1], **Yilin Zhang**[1],
**Ziyu Guan**[1], **Wei Zhao**[1*], **Fangyuan Zhao**[2], **Xin Yang**[1]
[1]School of Computer Science and Technology, Xidian University, China
[2]School of Computer Science and Technology, Xi'an Jiaotong University, China
{chenhaishun, 24031212287, ylzhang_3, xinyang}@stu.xidian.edu.cn
{cxu@, zyguan@, ywzhao@mail.}xidian.edu.cn
zfy1454236335@stu.xjtu.edu.cn

## Abstract

Multi-view evidential learning harnesses diverse data sources to improve prediction performance and provide reliable uncertainty estimates. Recent advances have primarily focused on optimizing evidence fusion strategies, assuming that the evidence extracted from each view is naturally reliable for downstream integration. However, our empirical analysis reveals that samples tend to be assigned biased evidence to support data-rich classes, thereby rendering unfair uncertainty estimations. This motivates us to delve into a new Biased Evidential Multi-view Learning (BEML) problem. To this end, we propose Fairness-Aware Multi-view Evidential Learning (FAML) method to rectify biased evidence learning. Specifically, FAML introduces the training-trajectory-based adaptive prior into the construction of Dirichlet parameters, flexibly calibrating the initial support evidence assigned to each class during training. Furthermore, we incorporate a fairness constraint as a regularization term to alleviate bias in the evidence. In the multi-view fusion stage, we propose an opinion alignment mechanism to mitigate view-specific bias across views, thereby encouraging the integration of consistent and mutually supportive evidence. Theoretical analysis shows that FAML effectively achieves less biased evidence allocation. Extensive experiments on real-world multi-view datasets demonstrate the superiority of our FAML, in terms of prediction performance and uncertainty estimation.

## 1 Introduction

Multi-view learning leverages complementary information from multiple views to improve model performance, providing significant advantages to real-world applications (Hu et al., 2022; 2023). For example, autonomous driving systems integrate data from multiple sensors to accurately perceive the driving environment (Cheng et al., 2024). However, in these high-stakes applications, it is essential for models to reliably represent the uncertainty associated with their predictions (Duan et al., 2024).

To address this practical limitation, a series of multi-view evidential learning (MVEL) methods based on Evidential Deep Learning (EDL)(Sensoy et al., 2018) have been proposed to provide classification predictions alongside the corresponding uncertainty. In the prevailing MVEL paradigm, evidence is collected separately from each view in accordance with Subjective Logic (SL) (Jøsang, 2016) and then aggregated to parameterize a Dirichlet distribution initialized with a non-informative uniform prior. Following this line of thought, many subsequent studies focus on refining the evidence-level fusion of multiple views to better handle real-world challenges such as inter-view conflict (Xu et al., 2024a; Lu et al., 2025) and low-quality views (Xu et al., 2024b; Liu et al., 2025c).

Although these methods demonstrate promising performance, most existing studies implicitly assume that the evidence extracted from each view is inherently impartial and reliable for downstream fusion. However, this assumption is often violated in real-world tasks, where data distributions are

---
*Corresponding author.

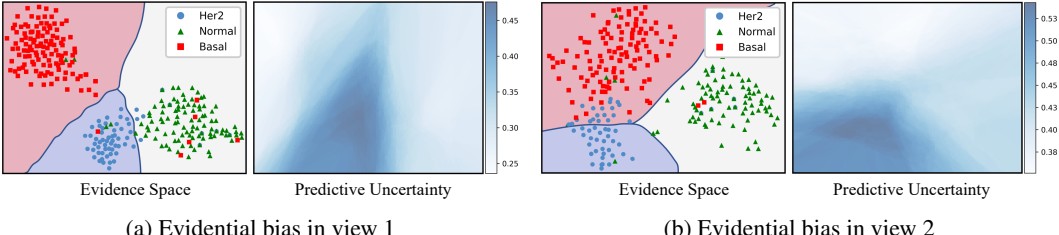

Figure 1: Visualization for the classification study and associated uncertainty on the BRCA dataset. To evaluate the predictive uncertainty across the feature manifold, we employ a dense sampling strategy to generate the test set. For both views, the left subfigures delineate the decision regions (shaded areas) overlaid with training instances (in points). The right subfigures depict the corresponding predictive uncertainty with a darker color indicating that the prediction is more uncertain.

often imbalanced due to inherent biases in the collection process(Chen et al., 2025). As a result, the probability of assigning more evidence is significantly higher for data-rich classes than for data-poor ones. More intuitively, as shown in Figure 1, an empirical analysis on the imbalanced real-world multi-view dataset Breast Invasive Carcinoma (BRCA) (Shea et al., 2020) highlights the unfairness in the allocation of evidence. We observe that for samples from the data-poor class (Her2), the supporting evidence is often largely assigned to the Normal class in the first view and to the Basal class in the second view. The view-specific biased evidence learning process leads Her2 samples to collect strong supporting evidence for data-rich classes, making them prone to being confidently misclassified. In contrast, correctly classified Her2 samples typically receive insufficient supporting evidence, which leads the model to assign low confidence to these correct predictions. This phenomenon poses a new Biased Evidential Multi-view Learning (BEML) problem, where the allocation of supporting evidence is inherently unfair, resulting in unreliable and biased predictive uncertainty.

To address the aforementioned problems, we propose a Fairness-Aware Multi-view Evidential Learning (FAML) method to address the BEML problem. Distinct from previous methods that primarily focus on how to aggregate evidence obtained from each view, FAML reformulates the evidential multi-view learning problem from the perspective of fair learning. As shown in Figure 2, to eliminate evidential bias across different classes, we introduce the training-trajectory-based prior into the construction of Dirichlet parameters, adaptively calibrating the prior support assigned to each class during training, thereby promoting balanced evidence allocation. With this adaptive design, the prior directly reflects class-wise performance, which naturally leads to a compensatory relationship: the worse a class performs, the larger the adaptive prior becomes. As the training proceeds, the adaptive prior would progressively converge to a fixed value used in standard Evidential Deep Learning. In addition, we also calculate the fairness degree according to the variance of class-wise evidence as an explicit constraint to regularize the evidence learning process. Considering that bias exhibits a view-specific pattern, we design an opinion alignment mechanism that minimizes the discrepancy of opinions between any pair of views, which ensures that different views align not only on what their predictions but also on how confident they are in their predictions.

The main contributions of this work are summarized as follows:

- We reveal a neglected but widespread issue in multi-view evidential learning, where the evidence learning process often exhibits implicit unfairness in practice.

- From the perspective of fairness, we propose a training-trajectory-based adaptive prior to deal with the issue of biased evidential multi-view learning that arises from class imbalance.

- We theoretically prove our adaptive prior increase the evidence margin for minority classes, and thereby guarantee improved generalization error bound by a factor of $\tilde{O}(1/\sqrt{\xi_k \Delta \beta_k})$.

- Extensive experiments on six real-world multi-view datasets demonstrate that FAML consistently outperforms existing methods, achieving fair and reliable uncertainty estimation.

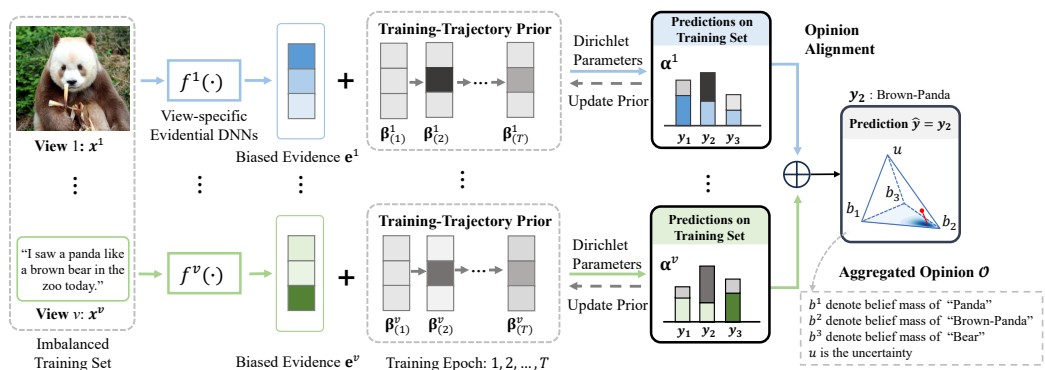

Figure 2: Illustration of FAML. It first employ view-specific evidential neural networks to construct opinions from each view. To mitigate evidential bias, the training-trajectory-based adaptive prior is introduced into the construction of Dirichlet parameters, flexibly calibrating the initial support evidence assigned to each class during training. Afterthat, an opinion alignment mechanism is then applied to mitigate view-specific bias across views. Finally, all view-specific opinions are integrated to reach a reliable and trustworthy prediction.

## 2 RELATED WORK

### 2.1 EVIDENCE THEORY.

As an arising single-forward-pass uncertainty estimation method, evidential deep learning (EDL) has attracted increasing research attention in recent years. Grounded in the subjective logic theory, EDL models predictive outputs as evidence for a Dirichlet distribution, offering reliable uncertainty estimation in various downstream tasks (Gao et al., 2024; Xia et al., 2024). The effectiveness of EDL critically depends on the evidence collection process, with recent studies exploring improvements such as evidence allocation refinement (Li et al., 2024), sample reweighting (Pandey & Yu, 2023), and advanced evidence functions(Shen et al., 2023; Yu et al., 2024) to enhance uncertainty quantification. However, most recent works still rely on a fixed uniform prior for the Dirichlet distribution(Chen et al., 2024), which may undermine the reliability in uncertainty estimation in practice. To address this gap, we introduce a training-trajectory-based adaptive prior, explicitly designed to mitigate evidence bias and enhance uncertainty reliability.

### 2.2 MULTI-VIEW EVIDENTIAL LEARNING.

To enhance the performance and reliability of models by integrating data from multiple views, multi-view evidential learning has attracted significant attention. Pioneering works like Trusted Multi-view Classification (Han et al., 2022b; Liu et al., 2022) employ the Dempster-Shafer theory (Shafer, 1992) to aggregate evidence from different views. Building on this research, a significant body of subsequent work has focused on making the fusion process more robust to real-world challenges. For example, some methods (Zhou et al., 2023; Xu et al., 2024a) focus on explicitly handling conflicting opinions between different views to achieve a more reliable decision. Others (Liu et al., 2024; 2025a;b) are designed to dynamically assign adaptive weights to each view, effectively reducing the influence of low-quality views during the fusion process. More recently, to improve transparency and reliance at both the evidence and the decision levels, TMCEK (Liang et al., 2025) introduce expert knowledge to identify critical evidences relevant to decision-making, enhancing the interpretability and robustness of multi-view learning. SAEML (Xu et al., 2025) explicitly normalizes the contribution of evidence from different views to ensure cross-view comparability, thus improving the reliance of multi-view fusion process. Despite their success, all of these methods focus on enhancing the consistency and comparability of evidence across views, while the fairness in the allocation of evidence in each view is entirely neglected, which is of great significance in safety-critical domains like medical diagnosis and autonomous driving. In contrast, our method improves fairness through an adaptive prior mechanism, thereby reducing the risks in final decision-making.

## 2.3 FAIRNESS IN MACHINE LEARNING.

Fairness in machine learning(Zhang et al., 2025; Zhou et al., 2026) often concerns whether a model performs consistently across different subpopulations of data, which are usually characterized by sensitive attributes such as gender, race, or age. Moving beyond sensitive-attribute fairness, there is a small but growing line of work focusing on imbalanced-label subpopulations, where quantity-imbalance between classes becomes the major source of unfairness. To alleviate such quantity-induced unfairness, some approaches resort to data pre-processing, aiming to create a balanced training set through resampling (Kim et al., 2023) or synthetic data generation (Marchesi et al., 2025). UMIX (Han et al., 2022a) is one of the most representative works in this line, which generates training samples by performing an uncertainty-aware mixup, thereby improving model generalization and fairness across groups. Besides, other works (Xinying Chen & Hooker, 2023; Yu et al., 2025) focus on facilitating a fairness notion into objective function to remove discrimination during the training process. GroupDRO (Sagawa et al., 2020) trains the model to minimize the worst-case loss over pre-defined groups, thus effectively mitigating discrimination in an alternated manner. In contrast to them, FAML adaptively adjusts priors based on training trajectories without subpopulation information, making it a general evidential learning framework in an intrinsic and flexible manner.

## 3 METHOD

### 3.1 PROBLEM FORMULATION

In the context of multi-view learning, each instance is characterized by multiple views. To clarify, consider a dataset $D = \{\{x_n^v\}_{v=1}^V, y_n\}_{n=1}^N$ with labels $y_n \in \{1, 2, \cdots, K\}$, where $x_n^v \in \mathbb{R}^{d_v}$ denotes the feature vector for the $v$-th view of the $n$-th instance. Multi-view evidential learning aims to integrate information from different views at the evidence level, and then aggregates the evidence to parameterize a Dirichlet distribution over class predictions, thereby enhancing both classification performance and reliability. However, in parctice, the learning process often suffers from quantity-induced bias, leading to a biased evidence allocation. To enforce fairness in multi-view evidential learning, the expected evidence assigned to the ground-truth class for each view should be class-invariant, which is formulated as:

$$\mathbb{E}_{(x,y=k)\in D}\left[e_k^v(x)\right] = \mathbb{E}_{(x,y=k')\in D}\left[e_{k'}^v(x)\right], \quad \forall k, k' \in \{1, 2, \ldots, K\} \tag{1}$$

The goal of BEML is to eliminate evidence bias in the learning process, and achieve fair evidence allocation and reliable uncertainty estimation.

### 3.2 FAIRNESS-AWARE MULTI-VIEW EVIDENCTIAL LEARNING

#### 3.2.1 VIEW-SPECIFIC EVIDENCE LEARNING.

EDL extends Subjective Logic (SL) to deep neural networks, providing the model with the ability to quantify uncertainty in its predictions. Specifically, for the specific view of an instance $(x_n^v)^{\text{I}}$, we employ Evidential DNNs to obtain view-specific evidence $\boldsymbol{e} = [e_1, \ldots, e_k]$, the evidence is then used to parameterize the Dirichlet distribution, where the $k$-th category Dirichlet distribution parameter $\alpha_k$ can be derived as:

$$\alpha_k = e_k + 1, \tag{2}$$

where the constant 1 represents a non-informative uniform Dirichlet prior, implicitly assuming that each class is equally probable in the absence of evidence. According to Subjective Logic theory, the belief masses $b_k$ and the uncertainty mass $u$ are defined as follows:

$$b_k = \frac{e_k}{S} = \frac{\alpha_k - 1}{S}, \; u = 1 - \sum_{k=1}^K b_k = \frac{K}{S}, \tag{3}$$

where $S = \sum_{k=1}^K \alpha_k$ is defined as the Dirichlet strength. To generate the final predictions, Subjective Logic models each view-specific prediction as multinomial opinions $w = (\boldsymbol{b}, u, \boldsymbol{a})$, and the

---

[1]In this section, we omit the super- and sub-scripts for clarity

projected probability distribution of multinomial opinions is given by:

$$p_k = b_k + a_k u = \frac{\alpha_k}{S} \quad \text{for} \quad k = 1, 2, \ldots, K. \tag{4}$$

Normally, the base rate $\boldsymbol{a} = [a_1, a_2 \ldots, a_k]$ is typically conformed to a uniform discrete distribution, $1/K$, assuming that no class preference is adopted.

While the above formulation enables uncertainty-aware multi-view prediction, it implicitly overlooks the impact of uniform Dirichlet prior assumptions on the fairness of evidence collection. In particular, an improper prior could dominate the Dirichlet posterior, especially for samples with less supporting evidence, undermining both the reliability and fairness of the multi-view learning process. To address this issue, we propose a training-trajectory-based prior as a form of adaptive regularization. Instead of employing a uniform Dirichlet prior in parameterizing the Dirichlet distribution, our approach adjusts the prior adaptively based on the learning history of each class. The training-trajectory-based prior for the $k$-th class in the training epoch $(t)^2$ is defined as follows:

$$\beta_k = \eta \cdot N_k / (\sum_{n:y_n=k} \kappa(y_n, f_\theta(x_n))),$$
$$\text{where} \quad \kappa(y_n, f_\theta(x_n)) = \begin{cases} 1, & \text{if } y_n = f_\theta(x_n). \\ 0, & \text{if } y_n \neq f_\theta(x_n). \end{cases} \tag{5}$$

where $N_k$ is the number of samples from the $k$-th class, and $\eta$ is the upweight factor that controls the strength of the prior. To construct the trajectory-based prior, we train the model using empirical risk minimization and record the prediction results $f_\theta(x_n)$ for each sample in the training epoch $t$ and then adaptively adjust the prior for each class based on its prediction. In this way, we update the Dirichlet concentration parameters for each class as follows:

$$\hat{\alpha}_k = e_k + \beta_k. \tag{6}$$

This adaptive prior exploits the learning trajectory of each class to adaptively calibrate their contribution in evidence learning process, thereby enhancing fairness and mitigating the evidential bias.

To further get an intuitive sense of the level of fairness among classes, we introduce a measure named Fairness Degree, which quantifies the variance of class-wise evidence allocation in each view.

**Definition 1 Fairness Degree**. Given the set of evidence vectors produced by Evidential DNNs across all samples, the fairness degree is defined as the variance of the class-wise average evidence:

$$\boldsymbol{f}(\{\bar{e}_k\}_{k=1}^K) = \text{Var}(\{\bar{e}_k\}_{k=1}^K) = \frac{1}{K} \sum_{k=1}^K (\bar{e}_k - \bar{e})^2, \tag{7}$$

where $\bar{e}_k$ represents the average evidence assigned to the ground-truth class for samples belonging to class $k$, and $\bar{e}$ is the global average of these class-wise average evidences.

Intuitively, this metric captures the fairness of evidence distribution across different classes. When the Fairness Degree is low, it indicates that each class receives relatively consistent amounts of evidence for their ground-truth predictions, suggesting fair treatment by the model. Conversely, a high Fairness Degree reveals that some classes consistently receive much more or much less evidence than others, signaling systematic bias in the evidence allocation process that could compromise the reliability of uncertainty estimation.

### 3.2.2 EVIDENTIAL MULTI-VIEW FUSION.

In this subsection, we focus on multi-view fusion based on view-specific opinions. Since each view provides its own prediction and associated uncertainty, it is necessary to aggregate these opinions to reflect the overall uncertainty of the final decision. To obtain a more comprehensive joint opinion, we taking into account the view-specific uncertainty to calculate aggregated evidence. Rather than treating each view equally, we propose a confidence-based evidence aggregation method as follows:

---

[2] we omit the subscript $t$ in the subsequent notation for simplicity

**Definition 2: Confidence-based Evidence Aggregation.** Given two opinions $\boldsymbol{w}^A = (\boldsymbol{b}^A, u^A, \boldsymbol{a}^A)$ and $\boldsymbol{w}^B = (\boldsymbol{b}^B, u^B, \boldsymbol{a}^B)$ over the same instance, let $e_k^A$, $e_k^B$ and $e_k$ represent the evidence for the $k$-th category from view $A$, view $B$, and the aggregated view, we have:

$$
\begin{aligned}
e_k &= \frac{(1 - u^A)e_k^A + (1 - u^B)e_k^B}{2 - u^A - u^B} \\
&= \frac{c^A}{c^A + c^B} e_k^A + \frac{c^B}{c^A + c^B} e_k^B,
\end{aligned}
\tag{8}
$$

where $c^A = 1 - u^A$ and $c^B = 1 - u^B$ represent the confidence of view $A$ and view $B$. Following Definition 2, we combine the evidence from views into a common aggregated representation. According to the SL theory, we can obtain the final multi-view joint opinion, and thus get the final probability of each class and the overall uncertainty (see appendix A.1 for detailed derivations).

To ensure different views to provide consistent evidence, we further introduce the Dissonance Degree of views in Definition 3, which is established according to opinion variance.

**Definition 3: Dissonance Degree of Views**. Given any view represented by the opinion $\boldsymbol{w^A}$, the variance of opinion for $k$-th class is derived from the Dirichlet PDF:

$$
\begin{aligned}
\mathrm{Var}\left(\alpha_k^v\right) &= \frac{\alpha_k \left(\sum_{k=1}^K \alpha_k - \alpha_k\right)}{\left(\sum_{k=1}^K \alpha_k\right)^2 \left(\sum_{k=1}^K \alpha_k + 1\right)} \\
&= \frac{p_k^v \left(1 - p_k^v\right) u^v}{K + u^v}.
\end{aligned}
\tag{9}
$$

then the dissonance degree between two views $A$ and $B$ is computed as:

$$
\boldsymbol{d}\left(\boldsymbol{w^A}, \boldsymbol{w^B}\right) = \sum_{k=1}^K \left|\mathrm{Var}\left(\alpha_k^A\right) - \mathrm{Var}\left(\alpha_k^B\right)\right|.
\tag{10}
$$

The Dissonance Degree measures the discrepancy in uncertainty estimation between views by comparing their Dirichlet variances, thereby reflecting cross-view consistency at the uncertainty level. Compared with traditional divergences such as Kullback-Leibler (KL) or Jensen-Shannon (JS) divergence on the expected Dirichlet probabilities, it captures second-order uncertainty rather than only first-order probability differences.

### 3.2.3 LOSS FUNCTION.

In this subsection, we will introduce the process of training evidential neural networks to obtain the multi-view joint opinion. As previously discussed, to eliminate evidence bias, we introduce the adaptive trajectory-based prior to obtain the concentration parameters of the Dirichlet distribution. For the instance $\{x_n^v\}_{v=1}^V$, the supervised loss is derived from the expected cross-entropy between the true label and the Dirichlet mean, which can be induced as:

$$
\begin{aligned}
L_{ace}\left(\hat{\boldsymbol{\alpha}}_n\right) &= \int \left[\sum_{k=1}^K -y_{nk} \log p_{nk}\right] \frac{\sum_{k=1}^K p_{nk}^{\hat{\alpha}_{nk}-1}}{B\left(\hat{\boldsymbol{\alpha}}_n\right)} d\boldsymbol{p}_n \\
&= \sum_{k=1}^K y_{nk}\left(\psi\left(S_n\right) - \psi\left(\hat{\alpha}_{nk}\right)\right),
\end{aligned}
\tag{11}
$$

where $\psi(\cdot)$ is the digamma function, $B(\cdot)$ is the $K$-dimensional multinomial beta function. Eq. 11 is the integral of the cross-entropy loss function on the simplex determined by $\hat{\boldsymbol{\alpha}}_n$. The above loss function ensures that the correct label of each sample generates more evidence than other classes, however, it cannot explicitly guarantee that the evidence assigned to the correct label is unbiased. To this end, we introduce an additional constraint term into the loss function, namely fairness loss:

$$
\mathcal{L}_{fc} = \mathrm{Var}\left(\{\bar{e}_k\}_{k=1}^K\right) = \frac{1}{K}\sum_{k=1}^K \left(\bar{e}_k - \bar{e}\right)^2, \quad \bar{e} = \mathrm{Avg}(\bar{e}_1, \dots, \bar{e}_K),
\tag{12}
$$

where $\bar{e}_k$ denotes the average evidence assigned to class $k$ within the current mini-batch, and $\bar{e}$ is the mean of these class-wise averages. To mitigate imbalance-induced bias, we employ a class-balanced learning strategy (Xia et al., 2022) defined by the following batch-wise loss:

$$\mathcal{L}_{acc} = \sum_{n=1}^{b} \frac{1}{N_{y_n}} \cdot \mathcal{L}_{ace}\left(\hat{\boldsymbol{\alpha}}_n\right) + \mu_t \cdot \mathcal{L}_{fc},$$

(13)

where $\mu_t = \min(1.0, t/T) \in [0, 1]$ is the balancing coefficient, $t$ is the index of the current training epoch, and $T$ is the annealing step. By increasing the influence of fairness constraint in loss, the optimization process avoids being dominated by biased evidence and stabilizes as training progresses.

In order to strengthen the reliability constraint among views during training, minimizing the degree of dissonance between opinions was adopted. The consistency loss for a mini-batch is given by:

$$\mathcal{L}_{con} = \sum_{n=1}^{b} \sum_{p=1}^{V} \sum_{p \neq q}^{V} \boldsymbol{d}\left(\boldsymbol{w_n^p}, \boldsymbol{w_n^q}\right).$$

(14)

To sum up, the overall loss function for a mini-batch can be calculated as:

$$\mathcal{L} = L_{\text{acc}} + \sum_{v=1}^{V} L_{\text{acc}}^{(v)} + \lambda \cdot \mathcal{L}_{\text{con}},$$

(15)

The model optimization is elaborated in Algorithm 1 (please refer to the Appendix A.3).

## 4 THEORETICAL STUDIES

In this section, we aim to establish the generalization guarantees of the FAML framework, particularly for minority classes. The analysis is grounded in the margin theory of statistical learning, which connects the margin distribution achieved on the training set to the model's generalization error. In the following, we first introduce the definition of *Evidence Margin*.

**Definition 4.1** (Evidence Margin). Let $h_\theta : \mathcal{X} \to \mathbb{R}_+^K$ denote a model, parameterized by $\theta$, that maps a sample $x_n$ to a $K$-dimensional non-negative evidence vector $e_n = [e_{n1}, \ldots, e_{nK}]$. The *evidence margin* $\rho_n(\theta)$ of $x_n$ is then defined as

$$\rho_n(\theta) = e_{nk} - \max_{j \neq k}\{e_{nj}\},$$

(16)

where $k$ is the ground-truth class label of $x_n$.

To formally establish the connection between such a sample-wise margin and the model's overall generalization ability, we now introduce a foundational result from statistical learning theory. The following lemma provides a general bound that relates the margin distribution achieved on the training set to the upper bound of the generalization error.

**Lemma 4.2** (Margin Generalization Bound (Bartlett & Shawe-Taylor, 1998)). *For a hypothesis space $\mathcal{H}$, given any margin $\rho > 0$, with probability at least $1 - \delta$, for all $h \in \mathcal{H}$, the true risk $R(h)$ satisfies:*

$$R(h) \leq \hat{R}_{S,\rho}(h) + C_1\sqrt{\frac{Complexity(\mathcal{H})}{N\rho^2}} + C_2\sqrt{\frac{\ln(1/\delta)}{N}},$$

(17)

*where $\hat{R}_{S,\rho}(h)$ is the proportion of samples in the training set $S$ with a margin less than $\rho$, $Complexity(\mathcal{H})$ is a complexity measure of the hypothesis space, and $C_1, C_2$ are constants.*

Lemma 4.2 shows that increasing the margin $\rho$ tightens the generalization bound via the $O(1/\rho)$ term. This allows us to use the change in $\rho$ as a bridge: by analyzing the effect of the FAML algorithm on $\rho$, we can explicitly determine its impact on the generalization error bound. Theorem 1.3 formalizes this analysis.

**Theorem 4.3.** *Let $h_\theta : \mathcal{X} \to \mathbb{R}_+^K$ be a model trained on a dataset $S$ of size $N$. For a minority class $k$ with imbalance ratio $\xi_k = N_{-k}/N_k \gg 1$, where $N_k$ and $N_{-k}$ are the number of samples belonging to class $k$ and other classes respectively, and under the condition that the adaptive prior $\beta_k$ increases when class $k$ performs poorly, minimizing the FAML objective $L(\theta)$ yields the following theoretical guarantees for the evidence margin $\rho_n(\theta) = e_{nk} - \max_{j \neq k} e_{nj}$:*

- *Margin Increase for Minority Class.* *The expected margin improvement for class $k$ is bounded below by:*

$$\mathbb{E}[\Delta\rho_n \mid y_n = k] \gtrsim \eta\psi'(S_{\hat{\alpha}}) \cdot \left( \xi_k \cdot \frac{\Delta\beta_k}{S_{\hat{\alpha}}} - O\left(\frac{1}{\hat{\alpha}_{nk}^2}\right) \right),$$

  *where $\eta$ is the learning rate, $S_{\hat{\alpha}} = \sum_j \hat{\alpha}_{nj}$, $\hat{\alpha}_{nj} = e_{nj} + \beta_j$, and $\Delta\beta_k > 0$ is the increase in the adaptive prior. For $\xi_k \gg 1$, the positive term dominates, ensuring a net margin increase.*

- *Tighter Generalization Bound.* *The increase in margin tightens the generalization bound. The effective margin $\rho_{\text{eff}}$ satisfies:*

$$\rho_{\text{eff}} \gtrsim \rho_0 \left( 1 + \lambda \cdot \xi_k \cdot \Delta\beta_k \right),$$

  *where $\rho_0$ is the baseline margin and $\lambda > 0$ is a scaling factor. This leads to an improved generalization error bound for the minority class by a factor of $\tilde{O}\left(1/\sqrt{\xi_k \Delta\beta_k}\right)$.*

Theorem 4.3 delineates how the adaptive prior enhances the evidence margin and tightens the generalization bound, with particular efficacy for minority classes. First, as captured by the lower bound on the expected margin improvement, the corrective mechanism scales with the class imbalance ratio $\xi_k$. This scaling property ensures that the model automatically allocates greater corrective effort to classes that are both under-represented and under-performing. Second, the generalization improvement factor of $\tilde{O}(1/\sqrt{\xi_k \Delta\beta_k})$ quantifies this behavior: it shows that greater data imbalance (larger $\xi_k$) coupled with stronger model self-feedback (larger $\Delta\beta_k$) leads to greater generalization gain. This result provides a theoretical foundation for the superior performance of our method on long-tailed distributions. We provide the proof in appendix A.2.

## 5 EXPERIMENT

### 5.1 EXPERIMENTAL SETUP

We briefly present the experimental setup here, including the experimental datasets, comparison methods and evaluation metrics.

**Dataset.** Following previous work (Xu et al., 2024a; Liu et al., 2024), we conducted experiments on six multi-view real-world datasets as follows: Handwritten, Animal, Scene15, YaleB, Caltech-101 and BRCA. For more detailed descriptions of the datasets, please refer to the appendix A.4.

**Compared Methods.** We compare the proposed method with the following methods. (1) Single-view evidential learning methods contain: **TLC** (Trustworthy Long-tailed Classification) (Li et al., 2022) combines class-imbalance classification and uncertainty estimation in a multi-expert framework. **I-EDL**( Information-based EDL) (Deng et al., 2023) is the SOTA evidential approach for uncertainty-aware learning in the single-view setting. **R-EDL** (Relaxed-EDL) (Chen et al., 2024) explores the relaxation of subjective logic assumptions to improve robustness. For single-view baselines, we report the results from the best-performing view. (2) Multi-view evidential learning methods contain: **TMC** (Trusted Multi-view Classification) (Han et al., 2022b) addresses the uncertainty estimation problem and produces reliable classification results. **ETMC** improves upon previous methods by adding a pseudo-view that concatenates multi-view features to provide complementary information. **CCML** (Consistent and Complementary-aware trusted Multi-view Learning) (Liu et al., 2024) explicitly decouples two types of evidence to enhance robustness to ambiguous views. **ECML**(Evidential Conflictive Multi-view Learning) (Xu et al., 2024a) is the SOTA method that proposed a fusion strategy for solving multi-view conflictive problems.

**Evaluation Metric.** Similar to (Li et al., 2022), we evaluate the performance of FAML with diverse metrics. Besides applying Classification Accuracy (ACC) to measure the classification prediction

accuracy in different class regions (including Head, Med, Tail regions), we also adopt Expected Calibration Error (ECE) (Naeini et al., 2015) to check the correspondence between predicted probabilities and empirical accuracy. Along with classification, we evaluate the performance on failure prediction using AUROC (McClish, 1989) and FPR-95 Liang et al. (2017), where the estimated uncertainties are employed to distinguish between correct and incorrect predictions.

Table 1: Performance comparison in terms of ACC(%) and ECE(%) on test sets, ± indicates the standard deviation for 5 random seeds, with the best mean scores highlighted in **bold**.

| Dataset | Method | ACC(%) ↑ | | | | ECE(%) ↓ | | | |
|---|---|---|---|---|---|---|---|---|---|
| | | All | Head | Med | Tail | All | Head | Med | Tail |
| Handwritten | TLC | 81.9 ± 1.1 | 98.0 ± 1.5 | 85.0 ± 1.9 | 76.5 ± 4.0 | 23.3 ± 1.3 | 38.1 ± 2.3 | 26.7 ± 1.8 | 22.6 ± 1.4 |
| | I-EDL | 78.9 ± 1.7 | 93.0 ± 1.4 | 82.0 ± 2.5 | 65.9 ± 3.8 | 22.2 ± 3.3 | 26.8 ± 1.4 | 24.3 ± 1.5 | 32.2 ± 3.5 |
| | R-EDL | 85.5 ± 1.2 | 99.5 ± 0.4 | 88.6 ± 0.8 | 72.5 ± 2.5 | 34.9 ± 2.4 | 28.9 ± 0.5 | 31.5 ± 1.2 | 35.2 ± 3.9 |
| | TMC | 84.2 ± 0.4 | 99.2 ± 0.2 | 89.0 ± 0.3 | 69.3 ± 1.0 | 28.9 ± 0.8 | 31.6 ± 0.5 | 23.1 ± 1.0 | 23.4 ± 3.5 |
| | ETMC | 90.2 ± 0.8 | 99.2 ± 0.0 | 92.2 ± 0.4 | 83.1 ± 2.3 | 26.4 ± 1.2 | 19.6 ± 0.3 | 17.8 ± 1.0 | 24.2 ± 2.3 |
| | CCML | 84.1 ± 2.4 | 99.0 ± 0.3 | 87.5 ± 1.7 | 70.2 ± 6.2 | 36.3 ± 1.8 | 34.2 ± 2.5 | 27.0 ± 1.4 | 36.3 ± 6.3 |
| | ECML | 79.4 ± 5.8 | **99.8 ± 0.3** | 80.8 ± 1.1 | 63.4 ± 1.4 | 31.9 ± 5.4 | 40.9 ± 3.2 | 34.0 ± 2.7 | 21.0 ± 4.2 |
| | Ours | **94.2 ± 0.3** | 98.3 ± 0.4 | **92.5 ± 0.5** | **92.5 ± 0.7** | **20.6 ± 1.6** | **25.1 ± 2.4** | **17.0 ± 3.3** | **20.2 ± 4.3** |
| Animal | TLC | 68.9 ± 1.8 | 86.3 ± 3.4 | 66.3 ± 7.0 | 55.6 ± 3.1 | 23.0 ± 5.0 | 37.3 ± 5.4 | 23.6 ± 3.8 | 21.6 ± 3.4 |
| | I-EDL | 64.0 ± 1.1 | 91.6 ± 1.9 | 67.9 ± 2.4 | 36.0 ± 3.8 | 20.8 ± 2.6 | 47.6 ± 6.5 | 25.2 ± 4.5 | 22.9 ± 3.1 |
| | R-EDL | 68.9 ± 1.7 | 90.0 ± 2.1 | 73.0 ± 4.4 | 46.4 ± 4.0 | 13.6 ± 1.2 | 36.0 ± 2.5 | 26.8 ± 2.3 | 23.2 ± 3.6 |
| | TMC | 64.8 ± 0.3 | 89.6 ± 0.4 | 64.8 ± 0.4 | 42.7 ± 0.9 | 16.8 ± 0.4 | 37.4 ± 0.6 | 22.3 ± 1.7 | 20.1 ± 3.1 |
| | ETMC | 64.5 ± 0.3 | 90.0 ± 0.4 | 64.8 ± 0.4 | 41.4 ± 0.8 | 12.2 ± 0.7 | 27.4 ± 0.6 | 18.7 ± 1.1 | 17.8 ± 1.0 |
| | CCML | 64.6 ± 0.7 | 89.8 ± 0.7 | 63.8 ± 0.4 | 42.9 ± 1.1 | 23.1 ± 1.1 | 46.3 ± 1.2 | 23.3 ± 1.6 | 19.8 ± 2.1 |
| | ECML | 65.5 ± 0.5 | 89.8 ± 0.7 | 65.3 ± 1.2 | 44.1 ± 0.3 | 20.5 ± 1.5 | 37.9 ± 1.1 | 23.9 ± 2.1 | 18.7 ± 1.9 |
| | Ours | **76.3 ± 0.4** | **91.6 ± 0.9** | **81.3 ± 0.7** | **57.9 ± 0.4** | **11.0 ± 1.0** | **13.7 ± 1.3** | **16.4 ± 1.7** | **17.6 ± 1.6** |
| Scene15 | TLC | 38.7 ± 0.4 | 65.9 ± 1.7 | 29.8 ± 2.0 | 20.5 ± 0.9 | 19.3 ± 2.0 | 38.6 ± 3.2 | 17.6 ± 5.6 | 17.8 ± 1.0 |
| | I-EDL | 25.6 ± 1.9 | 42.8 ± 5.9 | 18.6 ± 3.1 | 15.2 ± 3.0 | 16.6 ± 5.7 | 20.9 ± 5.6 | 21.3 ± 4.9 | 29.5 ± 4.2 |
| | R-EDL | 44.3 ± 1.5 | 71.4 ± 5.7 | 30.8 ± 3.4 | 30.8 ± 4.5 | 31.2 ± 2.1 | 46.4 ± 7.2 | 28.8 ± 2.5 | 25.8 ± 4.6 |
| | TMC | 42.0 ± 0.3 | 69.1 ± 1.3 | 30.5 ± 0.7 | 26.2 ± 1.2 | 11.8 ± 0.5 | 29.6 ± 1.5 | 13.7 ± 1.4 | 14.8 ± 1.8 |
| | ETMC | 50.2 ± 0.4 | 73.3 ± 0.9 | 42.4 ± 0.5 | 34.8 ± 0.3 | 22.7 ± 0.3 | 32.6 ± 1.0 | 16.9 ± 1.0 | 21.1 ± 2.5 |
| | CCML | 39.5 ± 0.9 | 63.9 ± 0.7 | 31.6 ± 0.7 | 23.1 ± 2.2 | 16.2 ± 0.5 | 28.6 ± 1.3 | 17.9 ± 1.0 | 17.5 ± 0.7 |
| | ECML | 37.3 ± 0.7 | 65.2 ± 1.4 | 25.7 ± 0.9 | 20.9 ± 1.5 | 17.4 ± 0.9 | 35.9 ± 1.5 | 13.2 ± 1.1 | 18.0 ± 1.1 |
| | Ours | **57.2 ± 1.3** | **75.2 ± 1.7** | **50.1 ± 1.6** | **46.3 ± 3.9** | **11.2 ± 3.6** | **18.8 ± 2.5** | **13.2 ± 2.0** | **12.4 ± 1.1** |
| YaleB | TLC | 73.1 ± 4.7 | 89.5 ± 7.6 | 80.0 ± 4.6 | 55.7 ± 1.8 | 29.9 ± 6.5 | 30.9 ± 3.8 | 37.8 ± 3.8 | 46.3 ± 2.0 |
| | I-EDL | 83.7 ± 1.4 | 86.7 ± 3.5 | 82.9 ± 3.8 | 82.1 ± 5.0 | 35.2 ± 3.6 | 48.2 ± 5.4 | 32.1 ± 5.8 | 49.5 ± 4.9 |
| | R-EDL | 85.4 ± 1.0 | 96.2 ± 1.9 | 83.8 ± 3.8 | 78.6 ± 5.9 | 47.8 ± 4.2 | 37.3 ± 2.5 | 39.3 ± 1.5 | 45.9 ± 3.6 |
| | TMC | 80.0 ± 0.6 | 98.1 ± 2.0 | 80.9 ± 0.4 | 64.3 ± 1.4 | 30.1 ± 2.1 | 52.9 ± 0.2 | 25.9 ± 0.7 | 39.9 ± 1.6 |
| | ETMC | 82.2 ± 0.4 | 99.1 ± 0.3 | 77.1 ± 1.9 | 72.9 ± 2.8 | 30.6 ± 2.4 | 55.7 ± 0.7 | 29.1 ± 3.5 | 38.8 ± 2.6 |
| | CCML | 78.0 ± 4.7 | 98.1 ± 2.3 | 77.1 ± 1.9 | 63.6 ± 1.4 | 39.6 ± 3.5 | 54.3 ± 3.8 | 35.7 ± 4.8 | 38.9 ± 1.4 |
| | ECML | 76.6 ± 1.1 | 90.5 ± 1.0 | 80.1 ± 2.1 | 62.9 ± 2.8 | 30.6 ± 2.8 | 51.1 ± 1.6 | 29.7 ± 1.9 | 45.1 ± 1.8 |
| | Ours | **88.3 ± 2.9** | **99.5 ± 0.2** | **81.9 ± 1.0** | **84.3 ± 1.7** | **10.9 ± 3.4** | **27.2 ± 1.1** | **15.2 ± 1.3** | **27.5 ± 1.6** |
| Caltech-101 | TLC | 74.2 ± 0.5 | 97.6 ± 0.9 | 72.9 ± 1.2 | 57.5 ± 1.3 | 26.2 ± 2.7 | 48.7 ± 3.0 | 29.6 ± 2.3 | 34.1 ± 3.1 |
| | I-EDL | 75.9 ± 1.4 | 99.1 ± 1.9 | 82.4 ± 1.1 | 53.6 ± 2.8 | 32.3 ± 2.6 | 56.9 ± 4.3 | 37.4 ± 3.7 | 30.5 ± 2.9 |
| | R-EDL | 70.0 ± 2.9 | 95.7 ± 1.7 | 77.6 ± 2.4 | 45.0 ± 3.7 | 24.8 ± 3.5 | 23.9 ± 4.4 | 34.7 ± 4.5 | 31.5 ± 3.8 |
| | TMC | 73.9 ± 2.2 | 95.2 ± 2.1 | 79.5 ± 4.4 | 53.6 ± 2.9 | 19.8 ± 2.3 | 25.8 ± 3.9 | 33.4 ± 3.9 | 31.9 ± 1.4 |
| | ETMC | 72.9 ± 1.1 | 95.2 ± 1.7 | 76.2 ± 3.2 | 53.6 ± 2.1 | 15.2 ± 2.4 | 21.1 ± 0.7 | 29.5 ± 3.5 | 30.4 ± 2.6 |
| | CCML | 71.9 ± 1.2 | 97.6 ± 2.1 | 71.9 ± 3.1 | 52.5 ± 1.8 | 29.4 ± 0.9 | 45.1 ± 2.6 | 31.9 ± 3.0 | 32.7 ± 1.9 |
| | ECML | 74.7 ± 0.8 | 97.1 ± 0.9 | 76.7 ± 6.6 | 56.4 ± 2.9 | 36.4 ± 1.7 | 45.2 ± 3.6 | 42.2 ± 4.1 | 34.3 ± 3.8 |
| | Ours | **83.6 ± 1.0** | **99.5 ± 0.9** | **88.1 ± 1.5** | **67.8 ± 2.8** | **14.1 ± 1.3** | **11.4 ± 4.2** | **17.2 ± 1.5** | **26.9 ± 3.4** |
| BRCA | TLC | 74.9 ± 2.4 | 89.6 ± 3.3 | 67.8 ± 2.4 | 46.3 ± 8.8 | 32.4 ± 3.9 | 38.2 ± 2.2 | 32.9 ± 4.1 | 50.9 ± 3.7 |
| | I-EDL | 76.1 ± 3.0 | 81.6 ± 4.7 | 71.8 ± 6.2 | 68.8 ± 4.3 | 44.0 ± 3.4 | 53.6 ± 3.4 | 35.0 ± 6.7 | 51.7 ± 4.9 |
| | R-EDL | 76.6 ± 2.5 | 89.7 ± 1.3 | 64.3 ± 3.7 | 62.5 ± 7.0 | 29.2 ± 3.3 | 24.5 ± 5.2 | 33.6 ± 4.9 | 49.7 ± 1.6 |
| | TMC | 74.3 ± 1.1 | **91.9 ± 2.3** | 62.5 ± 4.0 | 46.9 ± 7.2 | 30.1 ± 1.9 | 40.6 ± 3.7 | 29.9 ± 1.2 | 45.7 ± 3.7 |
| | ETMC | 77.1 ± 1.1 | 81.2 ± 1.6 | 71.4 ± 5.2 | 55.6 ± 6.2 | 31.8 ± 1.9 | 32.9 ± 3.3 | 32.9 ± 3.5 | 47.8 ± 6.4 |
| | CCML | 75.4 ± 1.5 | 90.8 ± 1.5 | 67.9 ± 4.1 | 46.8 ± 4.2 | 24.3 ± 4.0 | 34.1 ± 3.5 | 31.9 ± 5.2 | 50.1 ± 4.1 |
| | ECML | 77.1 ± 1.1 | 82.0 ± 3.2 | 69.4 ± 4.5 | 55.6 ± 5.5 | 40.7 ± 3.2 | 48.6 ± 5.0 | 32.6 ± 4.8 | 53.6 ± 5.4 |
| | Ours | **82.9 ± 2.4** | 82.7 ± 3.5 | **83.9 ± 5.2** | **81.2 ± 4.4** | **15.0 ± 1.4** | **13.5 ± 2.6** | **29.4 ± 4.3** | **44.3 ± 4.2** |

## 5.2 Experimental Results and Analysis

**Comparison with SOTA methods.** According to the results in Table 1, the following conclusion can be made: (1) Performance superiority: FAML demonstrates significant advantages over various advanced evidence-based methods on six real-world datasets. Taking the results on Caltech-101 dataset as an example, our method outperforms the second-best approach by 7.7% in overall accuracy and 10.3% in tail-region accuracy. While some methods, such as ECML, occasionally outperform FAML in isolated cases, FAML demonstrates more stable and consistent performance across all class regions, especially in tail class regions. (2) Effectiveness of Multi-View Fusion: Compared to single-view methods, the fusion of multiple views does not guarantee better performance, as biased evidence aggregation may even degrade accuracy. In contrast, FAML achieves more effec-

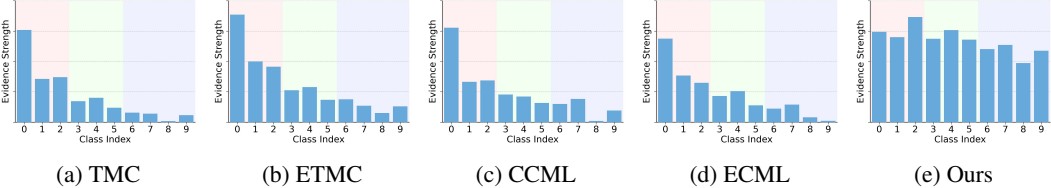

| (a) TMC | (b) ETMC | (c) CCML | (d) ECML | (e) Ours |

Figure 3: The average evidence strength of each class testing on Handwritten dataset. The shaded backgrounds denote different class regions based on the numbers of samples per class.

tive fusion by emphasizing informative views and aligning cross-view opinions, thereby delivering consistently superior results. (3) The results in terms of ECE reveal that our method maintains significantly lower ECE across all data region, which highlight the broad applicability and practical robustness of our method in uncertainty estimation. Moreover, we offer a more in-depth analysis for our FAML through the elaborate model analysis in the appendix A.6.

Table 2: AUROC and FPR-95 of uncertainty scores for identifying incorrect predictions.

| Data | Metric | TLC | I-EDL | R-EDL | TMC | ETMC | CCML | ECML | Ours |
|------|--------|-----|-------|-------|-----|------|------|------|------|
| Handwritten | AUROC(%) ↑ | 71.9 ± 2.3 | 61.5 ± 3.2 | 73.1 ± 0.8 | 78.4 ± 1.3 | 81.9 ± 0.9 | 70.8 ± 4.5 | 81.8 ± 0.8 | **85.7 ± 1.5** |
|  | FPR-95(%) ↓ | 78.5 ± 6.5 | 85.8 ± 3.6 | 90.9 ± 3.1 | 71.9 ± 0.8 | 66.5 ± 1.8 | 67.6 ± 3.0 | 76.8 ± 3.1 | **64.3 ± 2.8** |
| Animal | AUROC(%) ↑ | 76.0 ± 5.5 | 61.4 ± 7.8 | 76.2 ± 1.4 | 76.9 ± 0.8 | 79.4 ± 0.6 | 69.1 ± 0.7 | 76.9 ± 0.8 | **82.1 ± 0.6** |
|  | FPR-95(%) ↓ | 78.5 ± 3.1 | 80.6 ± 5.9 | 75.7 ± 2.9 | 74.4 ± 1.2 | 75.3 ± 1.2 | 82.1 ± 1.7 | 77.4 ± 3.3 | **75.3 ± 1.4** |
| Scene15 | AUROC(%) ↑ | 67.7 ± 2.0 | 56.3 ± 5.2 | 67.2 ± 1.5 | 70.8 ± 1.0 | 69.4 ± 2.9 | 63.7 ± 0.6 | 66.7 ± 0.5 | **71.2 ± 2.4** |
|  | FPR-95(%) ↓ | 82.6 ± 2.6 | 87.7 ± 1.3 | 88.9 ± 2.5 | 82.2 ± 2.8 | 80.8 ± 0.8 | 88.4 ± 0.7 | 84.5 ± 2.4 | **80.4 ± 2.6** |
| YaleB | AUROC(%) ↑ | 88.7 ± 1.5 | 71.8 ± 6.2 | 73.3 ± 3.8 | 70.8 ± 1.0 | 85.7 ± 1.3 | 88.5 ± 1.0 | 87.7 ± 1.1 | **90.3 ± 2.8** |
|  | FPR-95(%) ↓ | 61.0 ± 4.5 | 84.6 ± 7.3 | 79.9 ± 4.3 | 82.2 ± 2.8 | 82.2 ± 2.8 | 66.4 ± 2.6 | 64.7 ± 4.6 | **57.1 ± 3.6** |
| Caltech-101 | AUROC(%) ↑ | 75.8 ± 1.5 | 70.1 ± 5.4 | 72.6 ± 4.6 | 74.3 ± 2.4 | 73.8 ± 2.3 | 75.5 ± 1.2 | 74.9 ± 1.5 | **76.1 ± 1.4** |
|  | FPR-95(%) ↓ | 72.2 ± 6.3 | 82.1 ± 4.1 | 77.4 ± 5.5 | 77.1 ± 4.2 | 75.7 ± 4.4 | 75.0 ± 3.0 | 73.5 ± 4.9 | **69.8 ± 3.8** |
| BRCA | AUROC(%) ↑ | 74.8 ± 3.8 | 76.1 ± 2.3 | 73.7 ± 2.6 | 75.5 ± 3.2 | 76.6 ± 3.3 | 76.3 ± 1.5 | 77.1 ± 3.9 | **82.9 ± 3.9** |
|  | FPR-95(%) ↓ | 90.9 ± 3.2 | 95.1 ± 2.4 | 91.3 ± 3.4 | 83.7 ± 2.2 | 90.2 ± 2.8 | 84.7 ± 3.4 | 92.5 ± 3.0 | **75.6 ± 4.2** |

**Reliability of uncertainties.** Following prior work (Filos et al., 2019), we assess the reliability of uncertainty to ensure a thorough evaluation. Specifically, we perform the failure prediction task and employ AUROC and FPR-95 to quantify model's discriminate power in distinguishing incorrect predictions using uncertainty scores. As shown in Table 2, our method consistently achieves higher AUROC and lower FPR-95 across six datasets. These observations imply the reasonability of our model in estimating uncertainty, paving the way for reliable deep learning-driven high-risk applications in real-world settings. To comprehensively evaluate the uncertainty estimation, we further visualize the density distribution in terms of uncertainty in appendix A.6.

**Visualization of the distribution of evidence strength.** Real-world class imbalance often drives evidence to concentrate in data-rich classes while starving data-poor classes. To illustrate this phenomenon, Figure 3 reports the average evidence strength for each category on Handwritten dataset. We observe that compared methods tend to allocate much more evidence to head region classes while providing substantially less to tail region classes. In contrast, FAML achieves a relatively fair distribution of evidence strength by breaking the dependency between class sample size and evidence magnitude. Consequently, FAML is guaranteed to reduce the risk of overconfident misclassifications, thereby improving the overall reliability of predictions.

## 6 CONCLUSION

In this paper, we introduce a Fairness-aware multi-view evidential Learning method for addressing the biased evidential multi-view learning problem. FAML introduces a training-trajectory-based adaptive prior to construct Dirichlet distribution, which adaptively calibrates the Dirichlet parameters to mitigate view-specific evidential bias. Furthermore, we incorporate a fairness degree to constrain the view-specific evidence learning process, thus explicitly enhancing balanced evidence allocation. During the fusion stage, we incorporate an opinion alignment mechanism to guide the formation of the final fused opinion. Extensive experiments on real-world datasets confirm the effectiveness of FAML on prediction performance and reliable uncertainty estimation.

## ACKNOWLEDGMENTS

This work was supported in part by the National Natural Science Foundation of China under Grants 62425605, 62133012, 62472340, and 62373300, in part by the Natural Science Basic Research Program of Shaanxi (Program No. 2025JC-QYXQ-040), and in part by the Shaanxi Province outstanding Youth Science Foundation Project under Grants 2025JC-JCQN-074, and in part by the Key Research and Development Program of Shaanxi under Grants 2024CY2-GJHX-15, 2022ZDLGY01-10, and 2025CY-YBXM-041.

## ETHICS STATEMENT

This work adheres to the ICLR Code of Ethics. In this study, no human subjects or animal experimentation was involved. All datasets used, including Handwritten, Animal, Scene15, YaleB, Caltech-101 and BRCA, were sourced in compliance with relevant usage guidelines, ensuring no violation of privacy. We have taken care to avoid any biases or discriminatory outcomes in our research process. No personally identifiable information was used, and no experiments were conducted that could raise privacy or security concerns. We are committed to maintaining transparency and integrity throughout the research process.

## REPRODUCIBILITY STATEMENT

We have made every effort to ensure that the results presented in this paper are reproducible. All code and datasets have been uploaded as supplement material.

The experimental setup, including training steps, model configurations, and hardware details, is described in detail in the appendix A.5. Additionally, the multi-view dataset we used in experiments are publicly available, ensuring consistent and reproducible evaluation results. We authors believe these measures will enable other researchers to reproduce our work and further advance the field.

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

# A APPENDIX

In the supplemental material:

- **A.1:** We describe the derivations of Confidence-based Evidence Aggregation.
- **A.2:** We provide a theoretical proof of effectiveness of the adaptive prior.
- **A.3:** We describe the detailed algorithm steps of the FAML method.
- **A.4:** We give detailed descriptions of the datasets.
- **A.5:** We explain the details of the implementation.
- **A.6:** We present the results of the additional experiment.
- **A.7:** We describe the potential limitations of FAML and future work.
- **A.8:** We describe the Usage of Large Language Models.

## A.1 DERIVATIONS OF CONFIDENCE-BASED EVIDENCE AGGREGATION

This section derives the fused belief mass and uncertainty based on the fusion evidence in Confidence-based Evidence Aggregation.

$$
\begin{aligned}
b_k{}^{A \circledast B} &= \frac{e_k{}^{A \circledast B}}{K + \sum_{k=1}^{K} e_k{}^{A \circledast B}} \\
&= \frac{\frac{K \cdot b_k{}^A \cdot \frac{\left(1-u^A\right)}{u^B} + K \cdot b_k^B \cdot \frac{\left(1-u^B\right)}{u^B}}{2-u^A-u^B}}{K + \sum_{k=1}^{K} \frac{K \cdot b_k^A \cdot \frac{\left(1-u^A\right)}{u^A} + K \cdot b_k^B \cdot \frac{\left(1-u^B\right)}{u^B}}{2-u^A-u^B}} \\
&= \frac{b_k^A \cdot \frac{\left(1-u^A\right)}{u^A} + b_k^B \cdot \frac{\left(1-u^B\right)}{u^B}}{1 - u^A + 1 - u^B + \sum_{k=1}^{K} \left( b_k^A \cdot \frac{\left(1-u^A\right)}{u^A} + b_k^B \cdot \frac{\left(1-u^B\right)}{u^B} \right)} \\
&= \frac{b_k^A \cdot \frac{\left(1-u^A\right)}{u^A} + b_k^B \cdot \frac{\left(1-u^B\right)}{u^B}}{1 - u^A + 1 - u^B + \frac{\left(1-u^A\right)^2}{u^A} + \frac{\left(1-u^B\right)^2}{u^B}} \\
&= \frac{b_k^A \cdot \frac{\left(1-u^A\right)}{u^A} + b_k{}^B \cdot \frac{\left(1-u^B\right)}{u^B}}{\frac{1}{u^A} - 1 + \frac{1}{u^B} - 1} \\
&= \frac{b_k^A \cdot \left(1-u^A\right) \cdot u^B + b_k^B \cdot \left(1-u^B\right) u^A}{u^A + u^B - 2u^A u^B}.
\end{aligned}
\tag{18}
$$

According to Eq.2 and Eq.3 , we have:

$$
\begin{aligned}
u^{A \circledast B} &= 1 - \sum_{k=1}^{K} b_k{}^{A \circledast B} \\
&= 1 - \frac{\sum_{k=1}^{K} b_k^A \cdot \left(1-u^A\right) \cdot u^B + \sum_{k=1}^{K} b_k^B \cdot \left(1-u^B\right) u^A}{u^A + u^B - 2u^A u^B} \\
&= 1 - \frac{\left(1-u^A\right)^2 \cdot u^B + \left(1-u^B\right)^2 u^A}{u^A + u^B - 2u^A u^B} \\
&= \frac{\left(2 - u^A - u^B\right) u^A u^B}{u^A + u^B - 2u^A u^B}.
\end{aligned}
\tag{19}
$$

According to Eq.(19), when combining a opinion $\boldsymbol{w}^B$ with the original opinion $\boldsymbol{w}^A$, the aggregated uncertainty mass $u$ decreases if $u^B < u^A$, increases if $u^B > u^A$, and unchanged if $u^A = u^B$.

A.2 THEORETICAL PROOF

**Theorem A.1.** *Let $h_\theta : \mathcal{X} \to \mathbb{R}_+^K$ be a model trained on a dataset $S$ of size $N$. For a minority class $k$ with imbalance ratio $\xi_k = N_{-k}/N_k \gg 1$, where $N_k$ and $N_{-k}$ are the number of samples belonging to class $k$ and other classes respectively, and under the condition that the adaptive prior $\beta_k$ increases when class $k$ performs poorly, minimizing the FAML objective $L(\theta)$ yields the following theoretical guarantees for the evidence margin $\rho_n(\theta) = e_{nk} - \max_{j \neq k} e_{nj}$:*

- *Margin Increase for Minority Class. The expected margin improvement for class $k$ is bounded below by:*

$$\mathbb{E}[\Delta \rho_n \mid y_n = k] \gtrsim \eta \psi'(S_{\hat{\alpha}}) \cdot \left( \xi_k \cdot \frac{\Delta \beta_k}{S_{\hat{\alpha}}} - O\left( \frac{1}{\hat{\alpha}_{nk}^2} \right) \right),$$

  *where $\eta$ is the learning rate, $S_{\hat{\alpha}} = \sum_j \hat{\alpha}_{nj}$, $\hat{\alpha}_{nj} = e_{nj} + \beta_j$, and $\Delta \beta_k > 0$ is the increase in the adaptive prior. For $\xi_k \gg 1$, the positive term dominates, ensuring a net margin increase.*

- *Tighter Generalization Bound. The increase in margin tightens the generalization bound. The effective margin $\rho_{\text{eff}}$ satisfies:*

$$\rho_{\text{eff}} \gtrsim \rho_0 \left( 1 + \lambda \cdot \xi_k \cdot \Delta \beta_k \right),$$

  *where $\rho_0$ is the baseline margin and $\lambda > 0$ is a scaling factor. This leads to an improved generalization error bound for the minority class by a factor of $\tilde{O}\left( 1/\sqrt{\xi_k \Delta \beta_k} \right)$.*

*Proof.* We provide a detailed proof by analyzing the gradient dynamics of the FAML objective and their aggregation under class imbalance.

**Part 1: Gradient Analysis of the FAML Loss**

Consider a training sample $(x_n, y_n = k)$ with evidence vector $\mathbf{e}_n = h_\theta(x_n)$. The FAML loss for this sample is:

$$L_n(\theta) = \psi\left( \sum_{j=1}^K \hat{\alpha}_{nj} \right) - \psi(\hat{\alpha}_{nk}), \quad \text{where } \hat{\alpha}_{nj} = e_{nj} + \beta_j. \tag{20}$$

We first compute the gradient with respect to the correct class evidence $e_{nk}$:

$$
\begin{aligned}
\frac{\partial L_n}{\partial e_{nk}} &= \frac{\partial}{\partial e_{nk}} \left[ \psi\left( \sum_{j=1}^K \hat{\alpha}_{nj} \right) - \psi(\hat{\alpha}_{nk}) \right] \\
&= \psi'\left( \sum_{j=1}^K \hat{\alpha}_{nj} \right) \cdot \frac{\partial}{\partial e_{nk}} \left( \sum_{j=1}^K \hat{\alpha}_{nj} \right) - \psi'(\hat{\alpha}_{nk}) \cdot \frac{\partial \hat{\alpha}_{nk}}{\partial e_{nk}} \\
&= \psi'(S_{\hat{\alpha}}) \cdot 1 - \psi'(\hat{\alpha}_{nk}) \cdot 1 \\
&= \psi'(S_{\hat{\alpha}}) - \psi'(\hat{\alpha}_{nk}),
\end{aligned}
$$

where $S_{\hat{\alpha}} = \sum_{j=1}^K \hat{\alpha}_{nj}$.

Since $\psi'(x)$ is the Trigamma function, which is strictly positive and monotonically decreasing for $x > 0$, and since $S_{\hat{\alpha}} > \hat{\alpha}_{nk} > 0$, we have:

$$\psi'(S_{\hat{\alpha}}) < \psi'(\hat{\alpha}_{nk}) \quad \Rightarrow \quad \frac{\partial L_n}{\partial e_{nk}} < 0. \tag{21}$$

Now consider a sample $(x_m, y_m = l)$ where $l \neq k$. The gradient with respect to $e_{mk}$ (the evidence for class $k$ on this non-$k$ sample) is:

$$\frac{\partial L_m}{\partial e_{mk}} = \frac{\partial}{\partial e_{mk}} \left[ \psi \left( \sum_{j=1}^{K} \hat{\alpha}_{mj} \right) - \psi(\hat{\alpha}_{ml}) \right]$$

$$= \psi' \left( \sum_{j=1}^{K} \hat{\alpha}_{mj} \right) \cdot \frac{\partial}{\partial e_{mk}} \left( \sum_{j=1}^{K} \hat{\alpha}_{mj} \right) - \psi'(\hat{\alpha}_{ml}) \cdot \frac{\partial \hat{\alpha}_{ml}}{\partial e_{mk}}$$

$$= \psi' (S_{\hat{\alpha}}^m) \cdot 1 - \psi'(\hat{\alpha}_{ml}) \cdot 0$$

$$= \psi' (S_{\hat{\alpha}}^m) > 0,$$

**Part 2: Quantitative Analysis of the $\beta_k$ Effect**

Using the asymptotic expansion of the Trigamma function for large $x$:

$$\psi'(x) = \frac{1}{x} + \frac{1}{2x^2} + \frac{1}{6x^3} - \frac{1}{30x^5} + O \left( \frac{1}{x^7} \right), \tag{22}$$

We can analyze the quantitative effects of increasing $\beta_k$.

For a class $k$ sample, the update magnitude for increasing $e_{nk}$ is:

$$\Delta e_{nk} \propto -\frac{\partial L_n}{\partial e_{nk}} = \psi'(\hat{\alpha}_{nk}) - \psi'(S_{\hat{\alpha}})$$

$$= \left[ \frac{1}{\hat{\alpha}_{nk}} + \frac{1}{2\hat{\alpha}_{nk}^2} + O \left( \frac{1}{\hat{\alpha}_{nk}^3} \right) \right] - \left[ \frac{1}{S_{\hat{\alpha}}} + \frac{1}{2S_{\hat{\alpha}}^2} + O \left( \frac{1}{S_{\hat{\alpha}}^3} \right) \right].$$

For a minority class $k$ with $\xi_k \gg 1$, it is empirically observed that $\beta_k \gg e_{nk}$, we therefore approximate $\hat{\alpha}_{nk} = e_{nk} + \beta_k \approx \beta_k$ in the subsequent analysis. Meanwhile, since each adaptive prior $\beta_j$ is updated independently, we treat the contribution from other classes $C = \sum_{j \neq k} \hat{\alpha}_{nj}$ as constant when analyzing the effect of increasing $\beta_k$. Thus:

$$\Delta e_{nk} \approx \frac{1}{\beta_k} - \frac{1}{\beta_k + C} + O \left( \frac{1}{\beta_k^2} \right) = \frac{C}{\beta_k(\beta_k + C)} + O \left( \frac{1}{\beta_k^2} \right) = O \left( \frac{1}{\beta_k^2} \right). \tag{23}$$

For a non-class $k$ sample, the suppression update for $e_{mk}$ is:

$$\Delta e_{mk} \propto -\psi'(S_{\hat{\alpha}}^m) = - \left[ \frac{1}{S_{\hat{\alpha}}^m} + \frac{1}{2(S_{\hat{\alpha}}^m)^2} + O \left( \frac{1}{(S_{\hat{\alpha}}^m)^3} \right) \right]. \tag{24}$$

When $\beta_k$ increases by $\Delta\beta_k$, the change in suppression magnitude is:

$$\delta(\Delta e_{mk}) \approx \frac{\partial}{\partial \beta_k} \left[ -\frac{1}{S_{\hat{\alpha}}^m} \right] \Delta\beta_k$$

$$= \frac{1}{(S_{\hat{\alpha}}^m)^2} \cdot \Delta\beta_k + O \left( \frac{1}{(S_{\hat{\alpha}}^m)^3} \right).$$

This represents a reduction in the suppression force acting on class $k$ evidence from non-$k$ samples.

**Part 3: Aggregation under Class Imbalance**

The total relief effect accumulated from all non-class $k$ samples is:

$$\text{Total Relief} \approx N_{-k} \cdot \eta \cdot \frac{\Delta\beta_k}{(S_{\hat{\alpha}}^m)^2}. \tag{25}$$

The total suppression effect from all class $k$ samples is:

$$\text{Total Suppression} \approx N_k \cdot \eta \cdot O\left(\frac{1}{\beta_k^2}\right). \tag{26}$$

**Key Observation:** The evidence margin change $\Delta\rho_n$ for a class $k$ sample consists of two non-negative components:

$$\Delta\rho_n = \underbrace{\Delta e_{nk}}_{\text{from } k\text{-samples}} + \underbrace{(-\Delta(\max_{j \neq k} e_{nj}))}_{\text{from suppression on non-}k\text{ samples}}. \tag{27}$$

Both components are non-negative due to the gradient analysis in Parts 1 and 2. To establish a conservative lower bound, we focus on the first component while acknowledging that the second component provides additional margin improvement.

The net effect per class $k$ sample can thus be bounded below as:

$$\mathbb{E}[\Delta\rho_n \mid y_n = k] \geq \frac{1}{N_k}(\text{Total Relief} - \text{Total Suppression})$$

$$= \frac{1}{N_k}\left[N_{-k} \cdot \eta \cdot \frac{\Delta\beta_k}{(S_{\hat{\alpha}}^m)^2} - N_k \cdot \eta \cdot O\left(\frac{1}{\beta_k^2}\right)\right]$$

$$= \eta \cdot \left[\xi_k \cdot \frac{\Delta\beta_k}{(S_{\hat{\alpha}}^m)^2} - O\left(\frac{1}{\beta_k^2}\right)\right].$$

Using the first-order approximation $\psi'(S_{\hat{\alpha}}^m) \approx 1/S_{\hat{\alpha}}^m$ for the lower bound derivation, we obtain:

$$\mathbb{E}[\Delta\rho_n \mid y_n = k] \gtrsim \eta \cdot \left(\xi_k \cdot \frac{\Delta\beta_k}{S_{\hat{\alpha}}} - O\left(\frac{1}{\hat{\alpha}_{nk}^2}\right)\right). \tag{28}$$

The $\gtrsim$ notation indicates that this is a conservative lower bound due to both the asymptotic approximation and the deliberate omission of the second non-negative margin component. For $\xi_k \gg 1$ and significant $\Delta\beta_k$, the first term dominates, ensuring:

$$\mathbb{E}[\Delta\rho_n \mid y_n = k] > 0. \tag{29}$$

**Part 4: Generalization Bound Improvement**

By the margin-based generalization bound shown in lemma 4.2, for any $\rho > 0$, with probability at least $1 - \delta$:

$$R(h) \leq \hat{R}_{S,\rho}(h) + O\left(\sqrt{\frac{\text{Complexity}(\mathcal{H})}{N\rho^2}}\right). \tag{30}$$

The effective margin $\rho_{\text{eff}}$ can be characterized through the harmonic mean:

$$\frac{1}{\rho_{\text{eff}}^2} \propto \mathbb{E}[\rho_n^{-2}]. \tag{31}$$

The margin improvement derived above implies:

$$\mathbb{E}[\rho_n^{-2} \mid y_n = k] \lesssim \frac{1}{\rho_0^2} \cdot \frac{1}{(1 + \lambda\xi_k\Delta\beta_k)^2}, \tag{32}$$

where $\rho_0$ is the baseline evidence margin before model updating, $\lambda$ is a positive factor that absorbs the learning rate $\eta$, the inverse of the evidence scale $S_{\hat{\alpha}}$, and the conversion ratio from the margin expectation to the effective margin. By combining Eq.31 and Eq.32, we have:

$$\rho_{\text{eff}} \gtrsim \rho_0 (1 + \lambda\xi_k\Delta\beta_k). \tag{33}$$

Substituting into the generalization bound yields the stated improvement of order $\tilde{O}(1/\sqrt{\xi_k\Delta\beta_k})$.

$\square$

### A.3 THE FAML FRAMEWORK ALGORITHM.

FAML runs in two stages. Stage-1 (warm-up): train each view-specific evidential network with a fixed uniform prior to stabilize evidence extraction and record prediction trajectories. Stage-2 (joint training): periodically derive class-wise adaptive priors from the recorded trajectories, update the concentration parameters accordingly, and optimize a class-balanced objective augmented with a fairness constraint (to balance evidence allocation) and a consistency regularization (to align opinions across views).

At inference, obtain opinions from all views and fuse them via confidence-based aggregation to produce the final prediction and its uncertainty. The overall procedure is summarized in Algorithm 1.

---

**Algorithm 1** Pseudocode for FAML.

---

1: **Input:** Multi-view dataset: $\{\{\mathbf{x}_n^v\}_{v=1}^V, y_n\}_{n=1}^N$, warm-up epochs, hyperparameter $\eta$ and $\lambda$;
2: **Initialize:** Initialize the parameters $\theta$ of FAML;
3: **Output:** networks parameters;
4: ——————————Training——————————
5: /*Stage-1 Warm-up Training of View-specific Evidential Networks /
6: **for** warm-up epochs **do**
7:     **for** $v = 1 : V$ **do**
8:         $\mathbf{e}^v \leftarrow$ evidential network output;
9:         Calculate Dirichlet parameters $\alpha^v$ by Eq. 2;
10:         Obtain view-specific loss calculated by Eq. 11;
11:     **end for**
12:     Obtain overall loss by summing losses calculated by all $\{\alpha^v\}_{v=1}^V$;
13:     Update the parameters $\theta$ by gradient descent with the loss from above;
14: **end for**
15: /*Stage-2 Joint Training with Adaptive Prior /
16: **while** not converged **do**
17:     **for** $v = 1 : V$ **do**
18:         $\mathbf{e}^v \leftarrow$ evidential network output;
19:         Save the prediction results $\{f_\theta(x_n)\}_{n=1}^N$ of the current epoch;
20:         Obtain adaptive prior $\beta^v$ by Eq. 5;
21:         Calculate Dirichlet parameters $\hat{\alpha}^v$ by Eq. 6;
22:         Obtain view-specific loss calculated by Eq. 12;
23:     **end for**
24:     Calculate aggregated evidence $\mathbf{e}$ by Eq. 8;
25:     Obtain overall loss $L$ by Eq. 15;
26:     Update the parameters $\theta$ by gradient descent with the loss from above;
27: **end while**
28: **return** networks parameters;
29: ——————————Test——————————
30: Output: the final prediction and its uncertainty estimation;
31: **for** $v = 1 : V$ **do do**
32:     $\mathbf{e}^v \leftarrow$ evidential network output;
33:     Calculate Dirichlet parameters $\alpha^v$ by Eq. 2;
34: **end for**
35: Calculate aggregated evidence $\mathbf{e}$ by Eq. 8;
36: Calculate uncertainty $u$ by Eq. 3;
37: Calculate prediction $\mathbf{p}$ by Eq. 4;
38: **return** the prediction $\mathbf{p}$ and its corresponding uncertainty $u$;

---

A.4    DATASET DETAILS

The Handwritten Dataset, Animal Dataset, Scene15 Dataset, YaleB Dataset, Caltech-101 Dataset and BRCA Datasets are composed of pre-extracted vectorized features.

- **Handwritten** (Asuncion et al., 2007): It consists of 2,000 digit images evenly distributed across ten classes (0–9), with 200 samples per class. Each sample is represented from six complementary perspectives: pixel averages (PIX) computed over $2 \times 3$ grids (240 dimensions), Fourier descriptors (FOU) of character outlines (76 dimensions), Profile correlations (FAC) features capturing structural patterns (216 dimensions), Zernike moments (ZER) that offer rotation invariance (47 dimensions), Karhunen-Love (KAR) coefficients for compact representation (64 dimensions), and morphological attributes describing coarse shape characteristics (6 dimensions).

- **Animal** (Lampert et al., 2013): It consists of 10,158 images from 50 categories. For feature representation, we adopt deep features extracted by DECAF and VGG19 both with 4,096 dimensions.

- **Scene15** (Fei-Fei & Perona, 2005): It contains 4,485 images drawn from fifteen scene categories, covering both indoor and outdoor environments. Each image is characterized by three complementary feature representations: GIST descriptors (20 dimensions) that capture global spatial structure, Pyramid Histogram of Oriented Gradients (PHOG, 59 dimensions) encoding local edge and shape information at multiple scales, and Local Binary Pattern (LBP, 40 dimensions) features reflecting texture patterns.

- **YaleB** (Georghiades et al., 2002): It is a benchmark facial recognition dataset containing 650 images from 10 subject categories. Each image is described by three complementary feature views: Intensity (2,500 dimensions), Local Binary Patterns (3,304 dimensions), and Gabor features (6,750 dimensions).

- **Caltech-101:** (Fei-Fei et al., 2004) It contains 101 object categories, from which we select the top 20 categories for evaluation. For each image, we use deep features extracted from DECAF and VGG19, both represented as 4,096-dimensional vectors.

- **BRCA** (Serra et al., 2015): This is a real-world biomedical dataset collected from TCGA for breast invasive carcinoma subtype classification based on the PAM50 scheme. It consists of 875 patients unevenly distributed across five subtypes: Normal (115), Basal (131), Her2 (46), LumA (436), and LumB (147), reflecting the natural class imbalance in clinical data. Each patient is described by three omics views: mRNA expression (1,000 dimensions), DNA methylation (1,000 dimensions), and miRNA expression (503 dimensions).

In our setting, the training data exhibits significant class imbalance, where the classes can be equally separated into head, medium(med) and tail regions based on the different numbers of samples. Following the setting in most existing works on class imbalance problem, the test set is constructed to be class-balanced. Specifically, we randomly select 20% of the samples as the testing set and use the remaining data to construct a revised version of the training set by sampling a subset that follows a Pareto distribution (Arnold, 2014). To ensure reproducibility and serve as a benchmark for the community, these data partitions will be publicly released. We describe the revised datasets used in the experiments in detail and summarize the datasets in Table 3.

Specifically, the Handwritten dataset contains 404 instances across 10 categories, with category sizes ranging from 11 to 160. The Animal dataset includes 1,065 samples from 50 categories, with the largest class containing 311 instances and the smallest only 3. The Scene15 dataset is reduced to 914 samples across 15 categories, where category sizes vary between 14 and 328. The YaleB dataset now consists of ten facial categories with per-class sizes between 3 and 52 images.The the Caltech-101 dataset is revised to include 20 categories, where the number of instances per category ranges from 19 to 638. In contrast, the BRCA dataset is a real-world biomedical dataset with naturally imbalanced subtype distributions, and thus we directly adopt it without additional resampling. More details are illustrated in Figure 7.

Table 3: Statistic information of revised multi-view datasets

| View | Handwritten | Animal | Scene15 | YaleB | Caltech-101 | BRCA |
|---|---|---|---|---|---|---|
| $V_1$ | Fou(76) | DECAF(4,096) | GIST(20) | Intensity(2,500) | Gabor(48) | mRNA(1,000) |
| $V_2$ | Fac(216) | VGG(4,096) | PHOG(40) | LBP(3,304) | WM(40) | DNA(1,000) |
| $V_3$ | Kar(64) | - | LBP(59) | Gabor(6,750) | CENTRIST(254) | miRNA(503) |
| $V_4$ | MOR(6) | - | - | - | HOG(1,984) | - |
| $V_5$ | ZER(47) | - | - | - | GIST(512) | - |
| $V_6$ | PIX(240) | - | - | - | LBP(928) | - |
| #Imbalance Ratio ($\xi_k$) | 160:11 | 311:3 | 328:14 | 52:3 | 638:19 | 436:46 |
| #Label | 10 | 50 | 15 | 10 | 20 | 5 |

## A.5 IMPLEMENTATION DETAILS

**Model Architectures and Training Configurations**: Our computational environment was Ubuntu 20.04, with 64 GB DDR4 RDIMM, 1x 16-core Intel Core i7-12900K CPU @ 3.20 GHz, and NVIDIA RTX 3090 GPU (24 GB memory). For vector-type datasets, view-specific evidence was extracted using fully connected networks with ReLU activations. The model is trained for 200 epochs using the Adam (Kinga et al., 2015) optimizer with L2-norm regularization. More specifically, FAML training involves across two stages: First Stage - Warm-up Training: To avoid the influence of inaccurate predictions at the early stage, each view-specific evidential network was trained using the Adam optimizer for a fixed number of empirically determined epochs, during which the prior was fixed to 1. Second Stage - Joint Training with Adaptive Prior: We update adjust the prior with predictions every five epochs to ensure training stability.

To reduce randomness from data partitioning and imbalanced dataset construction, we adopted the strategy proposed in (Shi et al., 2022), repeating all random operations five times with different seeds and reporting the mean and standard deviation of results. Two hyper-parameters were particularly critical for performance tuning. We performed a grid search to identify optimal settings for each dataset. A detailed sensitivity analysis of these hyper-parameters is provided in a later section, while the final hyper-parameter configurations for the six datasets are summarized in Table 4.

Table 4: Detailed hyperparameters for FAML running on various datasets.

| | Handwritten | Animal | Scene15 | YaleB | Caltech-101 | BRCA |
|---|---|---|---|---|---|---|
| Learning rate | 5e-3 | 5e-3 | 5e-2 | 1e-2 | 2e-3 | 1e-2 |
| Weight decay | 1e-5 | 1e-5 | 1e-5 | 1e-3 | 1e-4 | 1e-5 |
| Batch size | 256 | 256 | 128 | 256 | 256 | 256 |
| Optimizer | Adam | Adam | Adam | Adam | Adam | Adam |
| Warm-up epochs | 20 | 20 | 30 | 20 | 20 | 30 |
| Maximum Epoch | 200 | 200 | 200 | 200 | 200 | 200 |
| Lr-patience | 20 | 30 | 10 | 10 | 20 | 10 |
| Lr-factor | 0.5 | 0.5 | 0.5 | 0.5 | 0.5 | 0.5 |

A.6 ADDITIONAL EXPERIMENT RESULTS

In this section, we provide additional ablation studies and experimental results to facilitate a comprehensive understanding of the FAML framework. They include:

1. Cohen's Kappa coefficient evaluation.

2. Comparative results of FAML with other methods in noisy scenarios.

3. Fairness evaluation and quantitative analysis.

4. Analysis of key components in the FAML framework.

5. Ablation study of warm-up epochs.

6. Visualization of uncertainty distribution across different class regions.

7. Hyperparameter sensitivity analysis.

**Cohen's Kappa Coefficient Evaluation.** Beyond conventional accuracy metrics, we further employ Cohen's Kappa Coefficient (McHugh, 2012) to provide a more robust evaluation of model performance. Cohen's Kappa measures the agreement between prediction and ground-truth, while accounting for the agreement that could be expected by chance. It means that our evaluation is not only driven by raw accuracy but also accounts for potential random consistency, thereby offering a more robust evaluation of classification performance.The experimental results are shown in Table 5, FAML consistently achieves the best Cohen's Kappa score on all six datasets, demonstrating superior performance even after correcting for chance agreement.

Table 5: Cohen's Kappa Coefficient(%) on six datasets. The best results are highlighted in **bold**.

| Dataset | Handwritten | Animal | Scene15 | YaleB | Caltech-101 | BRCA |
|---|---|---|---|---|---|---|
| TLC | 85.1 ± 1.8 | 64.7 ± 2.2 | 34.4 ± 0.3 | 70.2 ± 5.2 | 72.5 ± 2.4 | 65.1 ± 3.8 |
| I-EDL | 76.5 ± 1.9 | 63.3 ± 1.1 | 20.2 ± 2.0 | 71.4 ± 3.7 | 71.9 ± 1.7 | 65.4 ± 4.3 |
| R-EDL | 87.6 ± 1.0 | 68.2 ± 1.7 | 49.3 ± 0.8 | 77.8 ± 2.3 | 73.8 ± 2.2 | 67.4 ± 4.6 |
| TMC | 84.2 ± 0.4 | 64.4 ± 0.6 | 37.8 ± 0.4 | 76.5 ± 0.6 | 72.5 ± 2.2 | 64.3 ± 1.6 |
| ETMC | 89.6 ± 0.5 | 63.7 ± 0.3 | 46.4 ± 0.4 | 68.2 ± 1.0 | 70.4 ± 1.2 | 63.5 ± 2.1 |
| CCML | 82.3 ± 2.6 | 63.9 ± 0.7 | 35.2 ± 0.9 | 76.8 ± 0.8 | 75.9 ± 0.5 | 63.4 ± 2.4 |
| ECML | 64.4 ± 2.8 | 64.8 ± 0.5 | 32.7 ± 0.8 | 73.9 ± 1.2 | 73.4 ± 0.9 | 63.6 ± 1.5 |
| FAML (Ours) | **91.7 ± 0.3** | **75.7 ± 0.6** | **51.2 ± 0.6** | **77.8 ± 1.5** | **82.9 ± 0.7** | **74.0 ± 3.7** |

**Comparative results of FAML with other methods in noisy scenarios.** To demonstrate FAML's robustness in handling noisy or uncertain data, we also conducted evaluations on five noisy multi-view datasets. Specifically, we applied Gaussian noise to 20% of the original views, varying the noise standard deviation from 0.5 to 1.0. FAML underwent five iterations of testing on each dataset with both mean values and standard deviations reported. The results shown in Table 6. From the results, we observe that all comparison methods fall short of maintaining robust performance under noisy inputs. In contrast, FAML achieves more stable and consistent performance across all evaluated datasets. Notably, on on the Handwritten dataset, FAML achieved a performance that was 4.2% higher than the second best method, and on the Animal dataset, it surpassed the next best by 6.9%.

Table 6: Comparing the performance of FAML with other advanced methods on five noisy datasets.

| Dataset | Handwritten | Animal | Scene15 | YaleB | Caltech-101 |
|---|---|---|---|---|---|
| TLC | 82.5 ± 2.2 | 52.0 ± 3.0 | 27.8 ± 1.2 | 70.9 ± 4.3 | 70.6 ± 2.2 |
| I-EDL | 83.3 ± 0.7 | 53.5 ± 2.1 | 20.0 ± 2.6 | 74.6 ± 2.7 | 60.9 ± 0.9 |
| R-EDL | 84.2 ± 1.2 | 55.7 ± 2.7 | 44.4 ± 1.6 | 70.9 ± 1.3 | 70.4 ± 3.2 |
| TMC | 84.0 ± 0.8 | 63.9 ± 0.9 | 31.7 ± 1.4 | 71.4 ± 1.2 | 71.4 ± 2.2 |
| ETMC | 87.6 ± 1.5 | 67.7 ± 1.3 | 35.3 ± 1.0 | 78.2 ± 2.3 | 74.4 ± 1.6 |
| CCML | 80.7 ± 1.6 | 63.1 ± 0.7 | 29.3 ± 1.9 | 78.6 ± 1.9 | 76.8 ± 1.3 |
| ECML | 78.3 ± 3.8 | 65.5 ± 1.2 | 26.3 ± 2.7 | 74.3 ± 1.3 | 73.3 ± 2.4 |
| FAML (Ours) | **91.8 ± 0.8** | **74.6 ± 1.8** | **47.0 ± 1.2** | **85.5 ± 2.5** | **80.7 ± 1.3** |

**Fairness Evaluation and quantitative analysis.** To fully demonstrate the advantages of our method and further verify its effectiveness, we conduct experiments to provide quantitative results on fairness. To this end, we introduce a new metric named Fairness Degree to measure evidential bias across classes. As shown in Table 7, we observe that FAML achieves a much lower Fairness Degree, demonstrating our method exhibits significant advantages on promoting model's fairness across all five datasets. We attribute this to the incorporation of adaptive prior in the training stage, which ensures the model automatically allocates greater corrective effort to discriminated classes, thus helps mitigate the biased evidential multi-view learning problem to some extent.

Table 7: Fairness evaluation of FAML with other advanced methods on five datasets.

| Data | TLC | I-EDL | R-EDL | TMC | ETMC | CCML | ECML | FAML(Ours) |
|---|---|---|---|---|---|---|---|---|
| Handwritten | 0.1377 | 0.3717 | 4.7281 | 0.7287 | 0.9829 | 0.7058 | 0.8407 | **0.1005** |
| Animal | 0.7171 | 0.7652 | 4.0228 | 1.0958 | 1.4691 | 0.9626 | 0.7934 | **0.4457** |
| Scene15 | 1.2554 | 0.4330 | 3.3339 | 1.7937 | 2.0732 | 1.9479 | 1.7697 | **0.2743** |
| YaleB | 0.8278 | 0.1554 | 1.1779 | 0.5648 | 0.3850 | 0.2086 | 0.3340 | **0.1511** |
| Caltech-101 | 0.5615 | 1.4387 | 4.0557 | 2.6468 | 1.1432 | 0.7479 | 1.1427 | **0.2372** |

**Analysis of key components in the FAML framework.** To thoroughly validate the effectiveness of the adaptive prior, fairness constraint, and consistency regularization, we construct a detailed ablation study on six datasets. The experimental results are shown in Table 8. Specifically, to verify the effectiveness of the training-trajectory-based adaptive prior, we compare the baseline model (without any components) with the model incorporating only the adaptive prior (AP). The results shows that adding the adaptive prior yields prominent improvements across all datasets. For instance, on the BRCA dataset, accuracy increases from 74.3% to 77.5%, while ECE decreases from 28.0% to 21.5%. This provides crucial insights on the effectiveness of our proposed trajectory-based prior adjustment strategy. The fairness constraint further improve performance when combined with the adaptive prior, which is particularly pronounced in the ECE metric. We attribute this improvement to the explicit regularization of evidence variance, thus promoting more reliable uncertainty estimation. In contrast, the consistency regularization exhibits varying degrees of effectiveness across datasets, being particularly beneficial for datasets with a larger number of views. A possible reason is that as the number of views increases, the potential for inter-view conflict and dissonance also grows. In such scenarios, explicitly enforcing consistency becomes more critical to guide the model towards a consistent and reliable fused opinion.

Table 8: Ablation study on six datasets, "✓" means FAML with the corresponding component, "-" means not applied. The best results are highlighted by boldface.

| Component | | | Handwritten | | Animal | | Scene15 | | YaleB | | Caltech-101 | | BRCA | |
|---|---|---|---|---|---|---|---|---|---|---|---|---|---|---|
| AP | $\mathcal{L}_{fc}$ | $\mathcal{L}_{con}$ | ACC ↑ | ECE ↓ | ACC ↑ | ECE ↓ | ACC ↑ | ECE ↓ | ACC ↑ | ECE ↓ | ACC ↑ | ECE ↓ | ACC ↑ | ECE ↓ |
| – | – | – | 85.7 | 32.9 | 65.1 | 22.9 | 50.3 | 29.8 | 81.4 | 25.6 | 73.8 | 26.3 | 74.3 | 28.0 |
| ✓ | – | – | 86.3 | 27.7 | 72.9 | 16.9 | 55.1 | 20.4 | 85.1 | 20.9 | 79.1 | 20.3 | 77.5 | 21.5 |
| ✓ | ✓ | – | 87.3 | 25.2 | 72.3 | 13.1 | 56.2 | 13.4 | 87.1 | 18.1 | 82.3 | 22.4 | 81.1 | 16.2 |
| ✓ | – | ✓ | 90.8 | 24.7 | 72.6 | 20.2 | 55.6 | 15.7 | 84.3 | 21.9 | 80.9 | 20.2 | 77.3 | 19.2 |
| ✓ | ✓ | ✓ | **94.2** | **20.6** | **76.3** | **11.0** | **57.2** | **11.2** | **88.3** | **10.9** | **83.6** | **14.1** | **82.9** | **15.0** |

**Ablation study of warm-up epochs.** To ensure stable training, we first warm up the view-specific evidential networks before introducing training-trajectory-based adaptive prior. The key hyper-parameter of the warm-up stage is the warm-up epochs. We ablate different values of this hyper-parameter and evaluate the effect of it on the performance of our method. We conduct a detailed ablation study on six datasets, the comparison results are shown in Figure 4. According to the experimental results, it is observed that the performance from our FAML without using warm-up phase (0 epochs) generally attain worse results. In contrast, as the warm-up epochs increases, we can find that the model performance gradually improves within 10–20 epochs, while it may degrade slightly after 20 epochs. These findings confirm that an appropriate warm-up epochs is essential for balancing initial network stabilization with subsequent training.

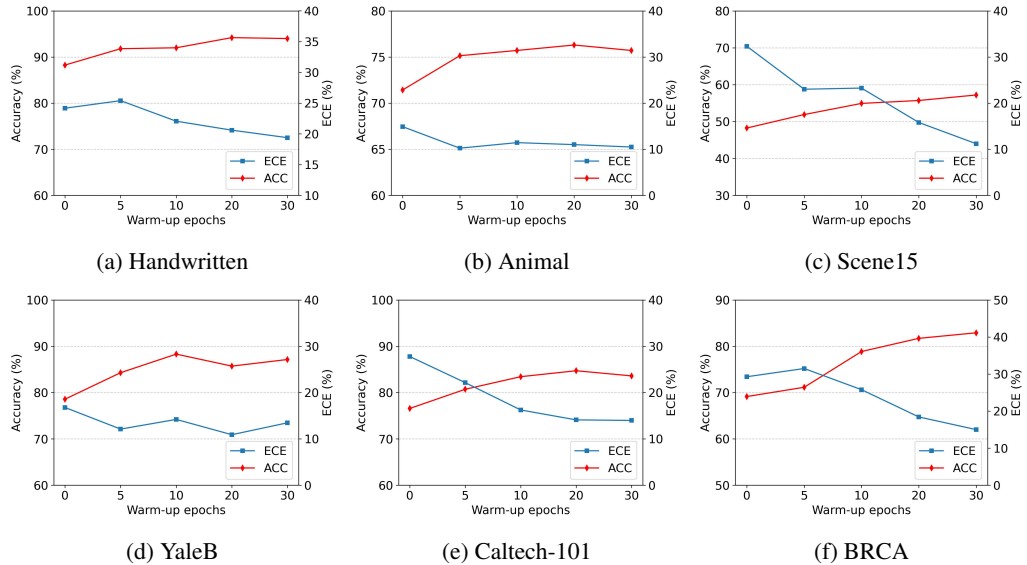

(a) Handwritten    (b) Animal    (c) Scene15

(d) YaleB    (e) Caltech-101    (f) BRCA

Figure 4: The effect of different warm-up epochs on performance.

**Visualization of uncertainty distribution across different class regions.** To comprehensively evaluate the uncertainty estimation, we visualize the density distribution for Animal dataset in terms of uncertainty. For comparison, we use our degraded baseline model that employs the same multi-view fusion strategy as FAML but without the fairness-aware components (adaptive prior, fairness constraint, and consistency regularization). Additionally, to ensure a fair comparison, the hyperparameters of this degraded baseline are kept strictly consistent with FAML. As shown in Figure 5, we reveal the following observations: (1) For the baseline method, the uncertainty estimates are relatively low for head classes, but increase substantially for medium and tail regions. The samples from data-rich classes tend to be assigned low uncertainty due to the biased evidence collection process, suggesting an unfair distribution of supporting evidence across classes. (2) In contrast, our method assigns lower and more compact uncertainty distributions across all regions, with notably reduced uncertainty for tail classes compared to the baseline. This indicates that FAML effectively mitigates evidence allocation bias through the adaptive prior and fairness constraint, delivering trustworthy predictions and demonstrating better fairness in uncertainty estimation across different class distributions.

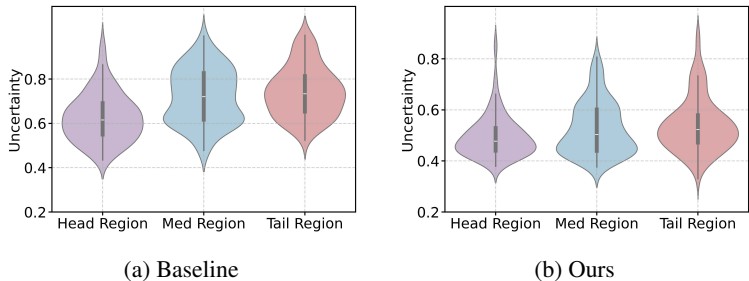

(a) Baseline    (b) Ours

Figure 5: Visualization of uncertainty estimation on the Animal dataset.

**Hyperparameter sensitivity analysis.** To further investigate the sensitivity of our proposed FAML model to its key hyperparameters, we analyze the impact of the upweight factor $\eta$, which controls the strength of the adaptive prior, and the hyperparameter $\lambda$ in consistency loss term, which determines the strength of consistency enforcement between views. As shown in Figure 6, for the upweight factor $\eta$, the performance exhibits a clear trend of first increasing and then decreasing. This indicates that large upweight factor may induce overconfidence in learning and ultimately degrade performance. For the hyperparameter $\lambda$, where $\lambda = 0$ means we do not consider the consistency loss

among opinions, the optimal range is $\lambda \in [1, 5]$. Our method exhibits robust performance across a wide range of parameter settings, highlighting its practical reliability and ease of deployment.

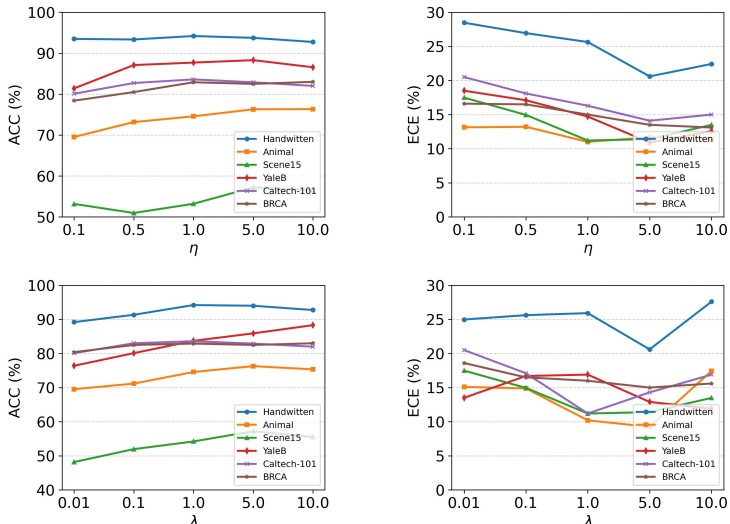

Figure 6: Sensitivity analysis of upweight factor $\eta$ and the hyperparameter $\lambda$ on datasets.

## A.7 POTENTIAL LIMITATIONS AND FUTURE WORK

Although the proposed method achieves excellent performance, it still has some potential limitations. (1) Our method focuses on supervised multi-view classification, which limits its applicability to scenarios where complete labeled data is available. Therefore, in future work, we will explore how to maintain fairness in evidence learning without explicit class labels, potentially through clustering-based approaches or contrastive learning mechanisms. (2) Our method provides an overall quantification of the predictive uncertainty, but does not disentangle the sources of uncertainty. It is promising to enhance uncertainty quantification by decoupling aleatoric and epistemic uncertainties under data imbalance, leading to more reliable interpretations of the outcomes and better understanding of when the model's predictions can be trusted. We consider these as important future works.

## A.8 THE USAGE OF LARGE LANGUAGE MODELS.

Large Language Models (LLMs) were employed to support the writing and refinement of this manuscript. Specifically, an LLM was utilized to improve readability, enhance clarity, and ensure grammatical accuracy across various sections of the text. The assistance included tasks such as rephrasing sentences, checking grammar, and improving the overall flow of presentation.

Importantly, the LLM was not involved in the formulation of research ideas, the design of methodologies, or the execution of experiments. All scientific concepts, analyses, and conclusions were developed independently by the authors. The role of the LLM was limited exclusively to linguistic refinement, without any contribution to the substantive scientific content.

The authors take full responsibility for the content of the manuscript, including any text generated or polished by the LLM. We have ensured that the LLM-generated text adheres to ethical guidelines and does not contribute to plagiarism or scientific misconduct.

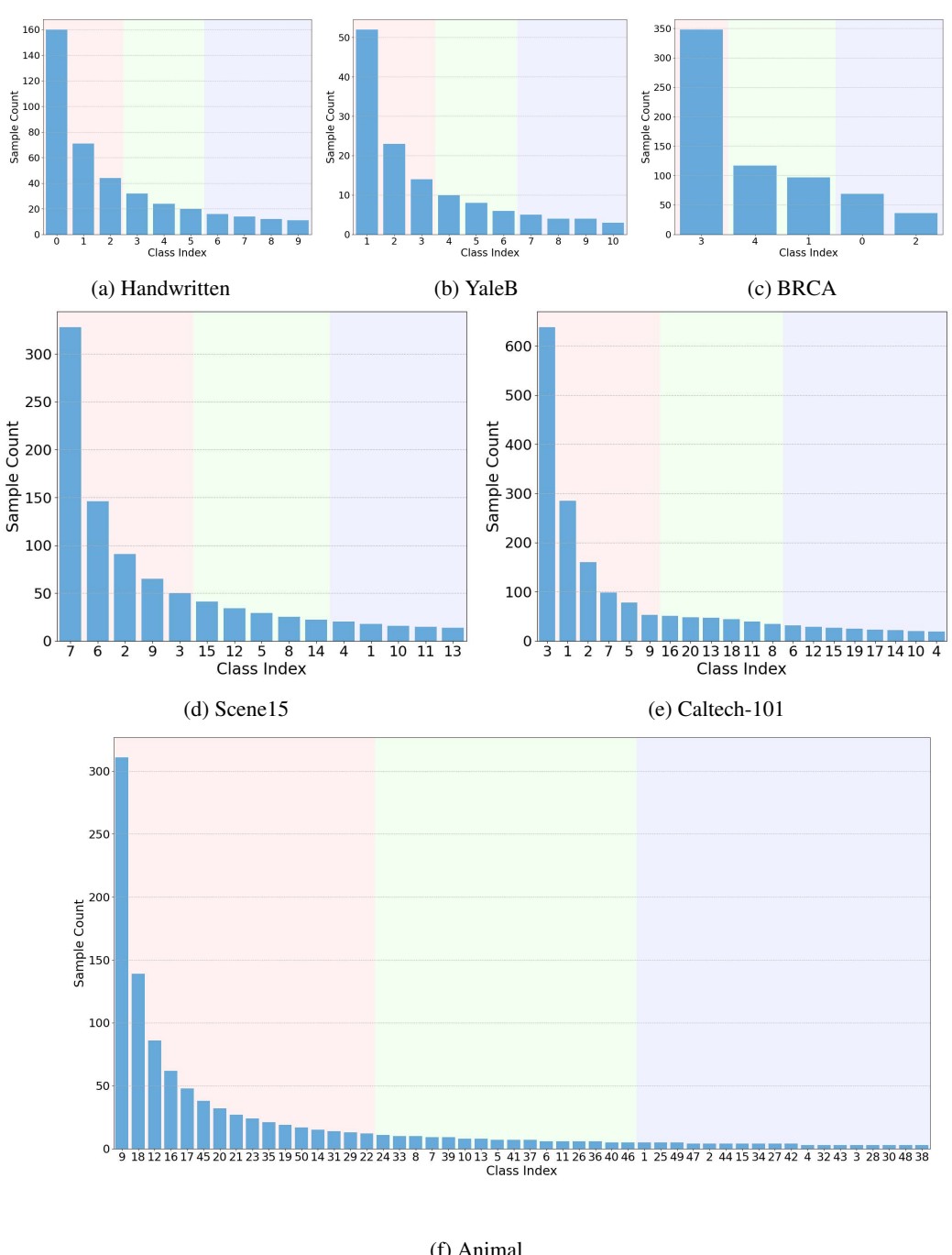

Figure 7: Train Dataset Distribution.The different background color regions correspond to head, medium, and tail categories, which are divided based on the numbers of samples in the dataset.

