# OpenReview forum: "Fairness-Aware Multi-view Evidential Learning with Adaptive Prior"
_ICLR.cc/2026/Conference — ICLR 2026 Poster_

### Official Review · Reviewer_Tpet · 2025-10-26

**Soundness:** 3
**Presentation:** 3
**Contribution:** 4
**Rating:** 8
**Confidence:** 4

**Summary:**

This paper addresses a critical bias in multi-view evidential learning: existing methods assume reliable view-specific evidence learning, but real-world data shows samples allocate more evidence to data-rich classes, harming uncertainty estimation reliability. To solve this newly defined Biased Evidential Multi-view Learning (BEML) problem, it proposes the Fairness-Aware Multi-view Evidential Learning (FAML) framework, which integrates three key components: a training trajectory-based adaptive prior (to calibrate bias), a class-wise evidence variance fairness constraint (to balance allocation), and an opinion alignment mechanism (to reduce cross-view bias). Theoretical analysis confirms FAML enhances evidence learning fairness, while experiments on 6 real-world datasets demonstrate FAML outperforms state-of-the-art methods in balanced evidence distribution, prediction performance, and uncertainty estimation reliability. The work also contributes by highlighting this implicit unfairness, proving the adaptive prior expands minority class evidence margins (with improved generalization error bounds), and validating FAML’s superiority.

**Strengths:**

-  The paper acutely captures a neglected yet prevalent issue in multi-view evidential learning: implicit unfairness in evidence allocation caused by differences in class data volume, which undermines the reliability of uncertainty estimation. This finding directly challenges the flawed assumption of existing methods that "view-specific evidence learning is inherently reliable," clarifies key optimization directions for future research, and reflects a deep understanding of practical pain points in the field.
- The proposed FAML method forms a closed-loop optimization from three dimensions, with strong targeting and novel design:
Adaptive Prior: Dynamically adjusted based on training trajectories, effectively calibrating evidence learning bias caused by class imbalance.
Fairness Constraint: Directly promotes balanced evidence allocation across different classes through class-wise evidence variance control.
Opinion Alignment Mechanism: Reduces view-specific bias during multi-view fusion, ensuring the integrated evidence is consistent and mutually supportive.
- The paper features both solid theoretical foundations and sufficient experimental validation, enhancing the persuasiveness of its research conclusions. First, it proves through derivation that the adaptive prior can expand the evidence margin for minority classes and provides an improved factor for the generalization error bound, offering mathematical support for the method’s effectiveness.
-  Extensive experiments on 6 real-world multi-view datasets not only verify FAML’s advantages in prediction performance but also demonstrate its ability to achieve fairer and more reliable uncertainty estimation, ensuring the universality of the results.

**Weaknesses:**

- The paper does not verify FAML’s performance under input noise (e.g., Gaussian noise on view features like GIST/HOG in Scene15), leaving unclear if its evidence fairness and uncertainty estimation hold for imperfect real-world data
- Though tested on datasets with 2-6 views, it lacks analysis on how view count changes (e.g., reducing Caltech-101’s 6 views to 3) affect FAML’s fairness (FD) and opinion alignment effectiveness

**Questions:**

- The paper updates the adaptive prior every 5 epochs starting from the 20th training epoch. Was there any preliminary experiment to confirm that a 5-epoch update interval is more suitable for maintaining training stability compared to other intervals (e.g., 3 or 10 epochs)?
- When defining the fairness degree (FD) based on class-wise evidence variance, did you observe how FD changes dynamically across training epochs (e.g., whether it drops faster in early or late stages)? And does this change trend correlate with the gradual increase of λ (the balancing coefficient) in the loss function?

---

> ### Author Response · Authors · 2025-11-22
> **Author response to Reviewer Tpet - Part 1**
>
> We would like to thank you for your valuable feedback, and we appreciate your recognition of our novel idea. We have carefully considered each of the concerns raised and have provided detailed responses below.
>
> > [W1] The paper does not verify FAML’s performance under input noise (e.g., Gaussian noise on view features like GIST/HOG in Scene15), leaving unclear if its evidence fairness and uncertainty estimation hold for imperfect real-world data.
>
> Thank you for your valuable questions. To demonstrate the robustness of FAML, we conducted additional experiments by introducing Gaussian noise into the data. Specifically, we applied Gaussian noise to 20% of the views, varying the noise standard deviation from 0.5 to 1.0. The comparison results are shown below.
>
>   | Method | HandWritten | Animal | Scene15 | YaleB | Caltech-101 |
>   | ------ | ----------- | ------ | ------- | ----- | ----------- |
>   | TLC    | 82.55       | 51.96  | 27.78   | 70.86 | 70.57       |
>   | IEDL   | 83.35       | 53.54  | 20.03   | 74.57 | 60.86       |
>   | REDL   | 84.20       | 55.71  | 44.44   | 70.86 | 70.43       |
>   | TMC    | 84.00       | 63.94  | 31.71   | 71.43 | 71.14       |
>   | CCML   | 80.65       | 63.14  | 29.27   | 78.57 | 74.43       |
>   | ECML   | 78.25       | 65.54  | 26.32   | 74.29 | 73.29       |
>   | Ours   | 91.75       | 74.63  | 46.98   | 85.5  | 80.71       |
>
> From the results, we observe that all comparison methods fall short of maintaining robust performance under noisy inputs. In contrast, FAML achieves more stable and consistent performance across all evaluated datasets, highlighting its ability to mitigate the influence of corrupted views through the adaptive calibration of the class-level Dirichlet prior. We have incorporated these results and discussions into the revised manuscript.
>
> > [W2] Though tested on datasets with 2-6 views, it lacks analysis on how view count changes (e.g., reducing Caltech-101’s 6 views to 3) affect FAML’s fairness (FD) and opinion alignment effectiveness
>
> We thank the reviewer for this insightful question. To verify the effectiveness of the FD constraint and opinion alignment under varying numbers of views, we have conducted experiments on the 6-view Caltech-101 and HandWritten datasets. The table compares Classification Accuracy (ACC) under different numbers of views.
>
>   | FD constraint | Opinion alignment | Caltech-101 (6 views) | Caltech-101 (3 views) |
>   | ------------- | ----------------- | --------------------- | --------------------- |
>   | √             |                   | 84.3                  | 49.3                  |
>   |               | √                 | 80.0                  | 41.1                  |
>   | √             | √                 | 87.1                  | 50.7                  |
>
>   | FD constraint | Opinion alignment | HandWritten (6 views) | HandWritten (3 views) |
>   | ------------- | ----------------- | --------------------- | --------------------- |
>   | √             |                   | 88.5                  | 72.2                  |
>   |               | √                 | 90.7                  | 68.3                  |
>   | √             | √                 | 93.6                  | 74.7                  |
>
> In summary, our experiments show that both components remain effective as the number of views changes. The FD constraint shows stable improvements under both view settings, indicating that its effectiveness is not sensitive to the number of available views. In contrast, opinion alignment becomes more helpful with a larger number of views, as more views tend to increase the variation in view opinions.

---

> ### Author Response · Authors · 2025-11-22
> **Author response to Reviewer Tpet - Part 2**
>
> > [Q1] The paper updates the adaptive prior every 5 epochs starting from the 20th training epoch. Was there any preliminary experiment to confirm that a 5-epoch update interval is more suitable for maintaining training stability compared to other intervals (e.g., 3 or 10 epochs)?
>
> Thank you for raising this question. We did conduct experiments to examine how different update intervals affect training stability. We compared update frequencies of 1,3, 5, and 10 epochs and found that a 5-epoch interval provides a better results. The comparison results on two example datasets are provided below.
>
>   | Prior Update Frequency/Dataset | 1    | 3    | 5    | 10   |
>   | ------------------------------ | ---- | ---- | ---- | ---- |
>   | HandWritten                    | 92.7 | 93.3 | 94.2 | 89.6 |
>   | Caltech-101                    | 81.4 | 82.1 | 83.6 | 82.8 |
>
> > [Q2] When defining the fairness degree (FD) based on class-wise evidence variance, did you observe how FD changes dynamically across training epochs (e.g., whether it drops faster in early or late stages)? And does this change trend correlate with the gradual increase of λ (the balancing coefficient) in the loss function
>
> Thank you for this thoughtful question. Following your suggestion, we visualized how the FD score changes during training on two example datasets, the anonymous links for the images are as follows:
>
> - https://i.postimg.cc/hjy2GQDh/FD-Scene15.png
> - https://i.postimg.cc/7LPcYk16/FD-Caltech101-20.png
>
> From this visualization, we observe that FD starts high and drops quickly in the early epochs, since the evidence across classes is still biased. As training goes on, FD becomes lower and much more stable. This pattern matches the gradual increase of the balancing coefficient, where a larger coefficient strengthens the fairness term and further reduces FD as training proceeds.
>
> -----
>
> We hope our rebuttal clarifies and addresses the your all concerns, and we also welcome any further constructive feedback you might provide to enhance our submission.

---

### Official Review · Reviewer_bZ2G · 2025-10-28

**Soundness:** 3
**Presentation:** 3
**Contribution:** 3
**Rating:** 6
**Confidence:** 3

**Summary:**

This paper introduces a novel and important problem: Biased Evidential Multi-view Learning (BEML). The authors demonstrate through empirical analysis that existing multi-view evidential learning methods tend to allocate more evidence to data-rich (majority) classes, leading to biased and unreliable uncertainty estimates for data-poor (minority) classes.

**Strengths:**

To address this, the paper proposes a Fairness-Aware Multi-view Evidential Learning (FAML) framework. The method has three core components:

An Adaptive Prior based on training trajectories, which dynamically adjusts the Dirichlet prior to provide more support to classes that are under-represented or poorly performing.

An explicit Fairness Constraint, which penalizes high variance in evidence allocation across different classes, encouraging a more balanced evidence distribution.

An Opinion Alignment mechanism, which minimizes the dissonance between opinions from different views during fusion to mitigate view-specific biases.

The authors provide strong theoretical grounding for their adaptive prior, proving that it increases the evidence margin for minority classes and tightens the generalization error bound. Comprehensive experiments on six real-world datasets show that FAML significantly outperforms state-of-the-art methods in terms of accuracy (especially for tail classes), calibration error (ECE), and uncertainty reliability (AUROC, FPR-95).

**Weaknesses:**

While this is an excellent paper, the following minor revisions could further strengthen its clarity and impact:

Strengthen the "Related Work" on Fairness and Subpopulation Shift: The core problem FAML solves—evidence bias due to class imbalance—is deeply connected to the broader fields of AI fairness and subpopulation shift. To better position the paper's contribution, the "Related Work" section should be expanded to include and discuss key works from this area. For instance, the authors should cite foundational work like GroupDRO (Sagawa et al., 2019), which formalized the goal of improving worst-group generalization in subpopulation shifts. More importantly, citing a paper like UMIX (Han et al., 2022), which explicitly links Uncertainty-Aware methods to solving Subpopulation Shift, would be highly relevant. Discussing these papers would allow the authors to clearly articulate FAML's unique contribution: while GroupDRO tackles the problem via the loss function and UMIX uses uncertainty to guide data augmentation, FAML introduces a novel approach by manipulating the evidential learning framework itself (via adaptive priors and fusion) to achieve fairness in a multi-view setting.

Intuition for the Adaptive Prior (Eq. 5): The paper should more explicitly state the key intuition behind Equation 5. The current description is accurate but subtle. The authors should clearly emphasize that this formula creates an inverse relationship: the worse a class performs (i.e., the fewer samples are correctly classified, the smaller the denominator), the larger the adaptive prior (Beta_k) becomes. This "compensatory" mechanism is the core of the idea and should be stated plainly.

Explicit Formulation of L_fc and mu schedule: In Section 3.2.3, the paper introduces the fairness loss L_fc, stating it is based on Definition 1. For absolute clarity, the paper should explicitly write out the final loss term (e.g., L_fc = Var(...)). Furthermore, the balancing coefficient mu is described as "gradually changing from 0 to 1." Please specify the exact schedule used (e.g., linear, exponential) to enhance reproducibility.

Motivation for Dissonance Degree (Eq. 10): The "Dissonance Degree" used for opinion alignment is a novel metric. The authors should add a brief sentence justifying this specific choice (sum of absolute differences in variance) over other, more traditional divergence measures (e.g., KL or JS divergence on the Dirichlet means/probabilities).

**Questions:**

As shown in weaknesses.

---

> ### Author Response · Authors · 2025-11-22
> **Author response to Reviewer bZ2G**
>
> We sincerely thank the reviewer for their valuable comments and suggestions, which have helped us strengthen our submission significantly. We are happy that the reviewer finds our work to address an important and novel research question, and our FAML provide strong theoretical grounding  for proposed adaptive prior. So we did our best to address the remaining concerns below.
>
> **Regarding the related work on Fairness and Subpopulation Shift:**
>
> Thank you for this valuable suggestion. Our work focuses on mitigating evidence bias caused by class imbalance in multi-view evidential learning, which s naturally related to the broader topics of AI fairness and subpopulation shift, where the goal is to prevent models from favoring certain groups or subpopulations.
>
> The suggested representative studies also lie on the paradigm of subpopulation shift, where the central idea is to improve robustness by reweighting loss functions (e.g., GroupDRO [1]) or guiding data augmentation through uncertainty (e.g., UMIX [2]). In contrast, FAML propose a more general evidential learning framework via adaptive priors  to achieve fairness in a multi-view setting.
>
> In the revised manuscript, we have cited and discussed the recommended paper, highlighting its relevance and clarifying how our work differs from or builds upon it.
>
> **Regarding the Intuition for the Adaptive Prior:**
>
> Thanks for the constructive suggestion. Our motivation is to calibrate the prior support assigned to each class during training, so that data-poor classes receive more initial evidential support. To achieve this, we design the prior to be dynamic rather than fixed, allowing it to adapt to the evolving training trajectory. With this adaptive design, the prior directly reflects class-wise performance, which naturally leads to a compensatory relationship: the worse a class performs, the larger the adaptive prior becomes. As training proceeds, the adaptive prior progressively converges to a fixed value used in standard Evidential Deep Learning, thereby making it a more general EDL framework to address practical challenges.
>
> In the revised manuscript, we incorporate this clarification to ensure that the motivation and intuitive behavior of the adaptive prior are clearly presented.
>
> **Regarding the Formulation Details:**
>
> Thank you for pointing out this. We would like to address your concerns with the following points:
>
> (1) The balancing coefficient is defined as $\mu_{t} = min (1.0, t / T)$, where t is the current training epoch and T is the annealing step. This corresponds to a linear schedule that gradually increases the strength of the fairness constraint from 0 to 1, which prevents the fairness constraint from influencing the optimization prematurely.
>
> (2) For completeness and reproducibility, we will explicitly include the closed-form expression of the fairness constraint in the revised manuscript. Specifically, based on Definition 1, the fairness loss is computed once per mini-batch as:
>
> $$
> L_{fc} = \operatorname{Var}(\({\hat{e}_k}\)\_{k=1}^K).
> $$
>
> where $\hat{e}_k$ denotes the batch-wise expected evidence for class $k$.
>
> In the revised manuscript, we have maked the formulation of fairness loss much clearer.
>
> **Regarding the Motivation for Dissonance Degree:**
>
> Thank you for this valuable comment. We would like to address your concerns with the following points:
>
> (1) Our primary motivation is to ensure that different views align not only on what their predictions but also on how confident they are in their predictions. The Dissonance Degree provides a principled metric for quantifying the level of agreement in uncertainty between any pair of views, thus reducing disagreement in their uncertainty patterns.
>
> (2) Traditional metrics like KL or JS divergence are typically applied to the expected probability vectors. While effective for aligning first-order probabilities, they fail to capture second-order uncertainty, which is essential for identifying how much evidential support each view assigns to the class.
>
> In the revised manuscript, we have added a brief explanation to clarify this design choice.
>
> [1] Distributionally Robust Neural Networks, ICLR 2020
>
> [2] Umix: Improving importance weighting for subpopulation shift via uncertainty-aware mixup, NIPS 2022
>
> ----
>
> We sincerely hope our rebuttal clarifies and addresses the your all concerns. We look forward to more discussions and will be happy to answer any further queries as well.

---

### Official Review · Reviewer_RwdE · 2025-10-30

**Soundness:** 3
**Presentation:** 3
**Contribution:** 3
**Rating:** 6
**Confidence:** 5

**Summary:**

The paper highlights an overlooked bias in multi-view evidential learning, where evidence allocation tends to favor data-rich classes, leading to unreliable uncertainty estimates. Instead of relying on fixed uniform priors like traditional EDL methods, it adaptively adjusts class priors based on training trajectories in a principled manner. Additionally, a fairness constraint on class-wise evidence allocation and an opinion-alignment regularization across views ensure the consistent allocation of evidence across views. Experiments on six multi-view datasets demonstrate superior region accuracy and ECE compared to baselines, with ablation studies confirming the contributions of each component.

**Strengths:**

1. This paper is well organized, and the proposed methodology is enlightening.

2. The motivation behind the paper is clear, and the theoretical analysis is complete.

3. The proposed method offers novel insights, particularly in using training trajectories to adjust class priors, thereby mitigating view-specific bias throughout the multi-view fusion process

4. The proposed method shows a clear performance improvement in a series of experiments.

**Weaknesses:**

1. In this paper, the notion of fairness seems to focus on balancing the evidence allocation across different classes, rather than addressing fairness in terms of sensitive attributes like race or gender in a broader sense.

2. Is this approach intended as a general framework? Can other trusted multi-view fusion methods also adopt similar strategies to improve model performance even on balanced datasets?

3. Some implementation details seem to be missing. For instance: How does the hyper-parameter $\mu$ change during training? and How is the metric ECE calculated in your experiments.

If the authors can address my questions, I am willing to increase my score.

**Questions:**

See Weaknesses.

---

> ### Author Response · Authors · 2025-11-22
> **Author response to Reviewer RwdE**
>
> Thanks for your insightful feedback and comments. We have carefully addressed each of the concerns raised with detailed responses below, and we hope these clarifications satisfactorily answer all questions.
>
> > [W1] In this paper, the notion of fairness seems to focus on balancing the evidence allocation across different classes, rather than addressing fairness in terms of sensitive attributes like race or gender in a broader sense.
>
> Thank you for raising this important point. Fairness in machine learning often concerns whether a model performs consistently across different subpopulations of data, which are usually characterized by sensitive attributes such as gender, race, or age. Moving beyond sensitive-attribute fairness, there is a small but growing line of work focusing on imbalanced-label subpopulations, where quantity-imbalance between classes becomes the major source of unfairness  [1,2,3]. Following this line, we define the fairness in evidential learning as the balanced allocation of evidential support across classes, aiming to achieve more reliable uncertainty estimations. We have added the corresponding explanation in the revised manuscript. Thank you again for providing important comments that helped improve the paper.
>
>
> > [W2] Is this approach intended as a general framework? Can other trusted multi-view fusion methods also adopt similar strategies to improve model performance even on balanced datasets?
>
> Thank you for your valuable comment. In response, we have conducted additional experiments on two example datasets and the results are as follows:
>
>   Scene 15:
>
>   | Fusion methods | Fixed Prior | Adaptive Prior | $\triangle$% |
>   | -------------- | ----------- | -------------- | ------------ |
>   | BCF            | 64.12       | 65.22          | 1.10         |
>   | ABF            | 60.98       | 62.21          | 1.23         |
>   | WBF            | 60.49       | 61.87          | 1.38         |
>
>   Animal:
>
>   | Fusion methods | Fixed Prior | Adaptive Prior | $\triangle$% |
>   | -------------- | ----------- | -------------- | ------------ |
>   | BCF            | 86.91       | 88.19          | 1.28         |
>   | ABF            | 87.80       | 89.22          | 1.42         |
>   | WBF            | 87.25       | 88.87          | 1.62         |
>
> In our experiment settings, evidence supporting all classes is fused using one of three fusion strategies: Belief Constraint Fusion (BCF) [4], Averaging Belief Fusion (ABF) [5], and Weighting Belief Fusion (WBF) [6]. From the above experiment results, it shows that the adaptive prior can be acting as a plug-and-play strategy to enhance model performance.
>
>
> > [W3] Some implementation details seem to be missing. For instance: How does the hyper-parameter change during training? and How is the metric ECE calculated in your experiments.
>
> Thank you for pointing out the missing implementation details. We provide the relevant clarifications below.
>
>  - The $\mu_{t} = \min (1.0, t / T) \in[0,1] $ is the balancing coefficient, where $t$ is the index of the current training epoch, and $T$ is the annealing step. By increasing the influence of fairness constraint in loss, the optimization process avoids being dominated by biased evidence and stabilizes as training progresses.
>
>  - We compute Expected Calibration Error (ECE) following the standard definition used in prior works. Specifically, predictions are partitioned into 10 equal-width confidence bins, and ECE is computed as
>   $$
>   \mathrm{ECE}=\sum_{m=1}^M \frac{\left|B_m\right|}{N}\left|\operatorname{acc}\left(B_m\right)-\operatorname{conf}\left(B_m\right)\right|
>   $$
> In our paper, the confidence is derived from the expected probability $\mathbb{E}\left[p_k\right]$ of the Dirichlet distribution, which is consistent with prior work [7]. We will add these details to the Implementation Section.
>
>
> [1] Properties of Fairness Measures in the Context of Varying Class Imbalance and Protected Group Ratios, TKDD 2024.
>
> [2] Fairness-aware Class Imbalanced Learning, EMNLP 2021.
>
> [3] Classes Are Not Equal: An Empirical Study on Image Recognition Fairness, CVPR 2024.
>
> [4] Trusted multi-view classification, ICLR 2021.
>
> [5] Reliable Conflictive Multi-View Learning, AAAI 2024.
>
> [6] Beyond Equal Views: Strength-Adaptive Evidential Multi-View Learning, MM 2025.
>
> [7] Better uncertainty calibration via proper scores for classification and beyond, NIPS 2022.
>
> ----
>
> We sincerely hope our rebuttal clarifies and addresses the reviewer's concerns. We look forward to more discussions and will be happy to answer any further queries as well.

---

### Official Review · Reviewer_yDqP · 2025-10-31

**Soundness:** 3
**Presentation:** 3
**Contribution:** 3
**Rating:** 6
**Confidence:** 5

**Summary:**

This paper addresses the issue of unreliable uncertainty estimation in multi-view evidence learning, which stems from biased evidence collection. The authors propose a framework called Fairness-Aware Multi-view Evidential Learning. This method uses a training-trajectory-based adaptive prior to calibrate the Dirichlet parameters, aiming to mitigate the evidence bias. The approach includes theoretical guarantees and is validated through experiments on six real-world datasets to demonstrate its performance.

**Strengths:**

1. The paper has a clear motivation and effectively solves the problems of biased evidence multi-view learning.

2. The paper offers a clear and well-grounded theoretical analysis that connects the adaptive prior design to margin theory, helping explain why the proposed approach could improve model's generalization.

3. The comparison experiments are comprehensive, including six representative multi-view datasets.

**Weaknesses:**

1. This work proposes an EDL-based multi-view classification method. However, the literature review for existing EDL-based multi-view methods is insufficient. The authors should provide a more comprehensive discussion of related work in this specific domain, such as, but not limited to, [1, 2].

2. The text in Figure 1 is too small, and there is no explanation of what the points, lines, and colors in the figure represent or why it is imbalanced. The blue in the legend of Figure 1 is different from that in the figure.

3. The phrase "most existing studies generally assume that view-specific evidence learning is inherently reliable" is ambiguous, especially the use of the word "reliable." What you want to convey is that the evidence learned from this view is unreliable, but it may lead readers to misunderstand that the view itself is unreliable.

4. Punctuation is also part of the formulas and needs to be added.

[1] Enhancing Testing-Time Robustness for Trusted Multi-View Classification in the Wild. CVPR 2025.

[2] Trusted multi-view classification with expert knowledge constraints. ICML 2025.

**Questions:**

1. What role did the opinion alignment play in promoting fairness? It seems irrelevant to fairness?

2. The paper positions fairness as a key design goal, yet there are no reported quantitative fairness evaluation metrics to determine this. Could the authors provide explicit metrics or quantitative analysis to support the fairness claims?

3. The degraded baseline model is introduced for the visualization. This baseline is described as FAML without the fairness-aware components. It is unclear if this is a re-run of an existing baseline (e.g. TMC) ? What are the hyper-parameters of the compared baseline?

4. In subjective logic, formulas rely on fixed priors to compute belief mass and uncertainty. When the prior becomes adaptive, do these formulations still hold as originally defined? Are the theoretical assumptions of subjective logic still satisfied after introducing the adaptive prior?

5. Could you discuss the robustness of FAML in the presence of potential noisy views. How does the adaptive prior perform in such scenarios?

6. Check for all possible typos in the manuscript. e.g., "bias exhibit view-specific pattern" should be "bias exhibits view-specific pattern" in Line 20.

---

> ### Author Response · Authors · 2025-11-22
> **Author response to Reviewer yDqP-Part 1**
>
> We would like to express our sincere gratitude for your valuable insights and suggestions on our work. We have tried to resolve your concerns as follows:
>
> > [W1] This work proposes an EDL-based multi-view classification method. However, the literature review for existing EDL-based multi-view methods is insufficient. The authors should provide a more comprehensive discussion of related work in this specific domain, such as, but not limited to, [1, 2].
>
> Thanks for the constructive suggestion. To strengthen the contribution and impact of our work, we have revised the related work section to include comprehensive discussion on existing EDL-based multi-view methods. In the revised section, we have emphasized that prior studies mainly focus on refining the integration rules of different views under complex sceneries while overlooking the fairness issues in evidence allocation, which is of great significance in safety-critical domains like  medical diagnosis and autonomous driving.
>
> > [W2] The text in Figure 1 is too small, and there is no explanation of what the points, lines, and colors in the figure represent or why it is imbalanced. The blue in the legend of Figure 1 is different from that in the figure.
>
> Thank you for your thoughtful feedback, which will undoubtedly improve the quality of our paper. We acknowledge that the current Figure 1 does not sufficiently convey the empirical motivation of our work. To make it easier for readers to grasp the underlying evidential bias revealed through our empirical analysis, we have added more details explanations on Figure 1 and enhance the clarifications of its motivation.
>
> > [W3] The phrase "most existing studies generally assume that view-specific evidence learning is inherently reliable" is ambiguous, especially the use of the word "reliable." What you want to convey is that the evidence learned from this view is unreliable, but it may lead readers to misunderstand that the view itself is unreliable.
>
> Thanks for highlighting this issue. We apologize for any ambiguous expression caused by our phrasing. In prior studies, the evidence captured from each view is always considered as impartial and trustworthy. However, the above assumption is often violated in real-world tasks, where the probability of allocating more evidence for data-rich classes is very high. To make this point clearer, we have refined the relevant descriptions and clarify the underlying concept in the revised manuscript.
>
> > [W4] Punctuation is also part of the formulas and needs to be added.
>
> Thanks for pointing out this issue. We have thoroughly checked the manuscript and will add or adjust punctuation in all formulas in the revised version.

---

> ### Author Response · Authors · 2025-11-22
> **Author response to Reviewer yDqP-Part 2**
>
> > [Q1] What role did the opinion alignment play in promoting fairness? It seems irrelevant to fairness?
>
> Thank you for this insightful question. We appreciate the opportunity to further  clarify the role that opinion alignment plays in promoting fairness within FAML. The key point is that opinion alignment introduces an explicit cross-view interaction, encouraging different views to produce consistent and mutually supportive evidence. Therefore, the opinion alignment can help model to reduce view-specific bias. We have incorporated this clarification into the revised manuscript.
>
> > [Q2] The paper positions fairness as a key design goal, yet there are no reported quantitative fairness evaluation metrics to determine this. Could the authors provide explicit metrics or quantitative analysis to support the fairness claims?
>
> We appreciate your comments on this issue. We authors agree that beyond visualizing the distribution of evidence strength, providing quantitative results on fairness is beneficial to support our fairness claims. To this end, we have conducted experiments on three example datasets with a new evaluation metric, Fairness Degree, to measure evidential bias across different classes. As shown in the table below, we observe that FAML achieves a much lower Fairness Degree, demonstrating our method exhibits significant advantages on promoting model's fairness across all five datasets. We attribute this to the incorporation of adaptive prior in the training stage, which ensures the model automatically allocates greater corrective effort to discriminated classes, thus helps mitigate the biased evidential multi-view learning problem to some extent. We have incorporated these results and discussions into the revised manuscript.
>
> | Methods/FD Degree | HandWritten | Animal | Scene15 |
> | ----------------- | ----------- | ------ | ------- |
> | TLC               | 0.1377      | 0.7171 | 1.2554  |
> | I-EDL              | 0.3717      | 0.7652 | 0.4330  |
> | R-EDL              | 4.7281      | 4.0228 | 3.3339  |
> | TMC               | 0.7287      | 1.0958 | 1.7937  |
> | CCML              | 0.7058      | 0.9626 | 1.9479  |
> | ECML              | 0.8407      | 0.7934 | 1.7697  |
> | Ours              | 0.1005      | 0.4457 | 0.2743  |
>
> > [Q3] The degraded baseline model is introduced for the visualization. This baseline is described as FAML without the fairness-aware components. It is unclear if this is a re-run of an existing baseline (e.g. TMC) ? What are the hyper-parameters of the compared baseline?
>
> Thank you for providing a detailed question. The "degraded baseline" is not a re-run of an existing method. Instead, it is an ablated version of our proposed FAML framework. Additionally, to ensure a fair comparison, the hyperparameters of this degraded baseline are kept strictly consistent with FAML, except for the parameters specific to the removed components. We will explicitly clarify these experimental settings in the revised manuscript. We have explicitly clarified these experimental settings in the revised manuscript.

---

> ### Author Response · Authors · 2025-11-22
> **Author response to Reviewer yDqP-Part 3**
>
> > [Q4] In subjective logic, formulas rely on fixed priors to compute belief mass and uncertainty. When the prior becomes adaptive, do these formulations still hold as originally defined? Are the theoretical assumptions of subjective logic still satisfied after introducing the adaptive prior?
>
> We thank the reviewer for this insightful question. In the standard SL framework, the Dirichlet parameters are often initialized with a fixed prior. We generalize this by replacing the fixed constant with an adaptive prior $\beta_k$, derived from the learning trajectory. Mathematically, the Dirichlet parameters are defined as (we omit the super-scripts for clarity):
> $$
> \hat\alpha_k=e_k+\beta_k.
> $$
> where $e_k$ is the evidence produced by the network and $\beta_k$ is the adaptive prior. Let $\beta_0 = \sum_{k=1}^K\beta_k $ denote the total prior strength. The Dirichlet strength is $S=\sum_{k=1}^K e_k+\beta_0$.
> Following standard subjective logic, belief and uncertainty are defined as:
> $$
> b_k=\frac{e_k}{S}, \quad u = \frac{\beta_0}{S}.
> $$
> It is easy to verify that the fundamental constraint of SL is strictly satisfied:
> $$
> u+\sum_{k=1}^K b_k=\frac{\beta_0}{S}+\sum_{k=1}^K \frac{e_k}{S}=\frac{\beta_0+\sum e_k}{S}=1.
> $$
> Actually, our core innovation lies in how this adaptive prior influences the Base Rate distribution. In SL, the expected probability $p_k$ is projected as:
> $$
> p_k=b_k+a_k \cdot u.
> $$
> where $a_k$ is the base rate. In our formulation, the adaptive prior effectively modifies this base rate to:
> $$
> a_k={\beta_k}/{\beta_0}.
> $$
> Specifically, by assigning larger priors $\beta_k$ to classes that exhibit poor training performance, we enable the uncertainty component ($a_k \cdot u$) to effectively supplement predictions for samples that lack sufficient evidence. In this way, the adaptive prior serves as a principled compensatory mechanism that strengthens support for  poorly learned classes while preserving all subjective-logic assumptions.
>
> > [Q5] Could you discuss the robustness of FAML in the presence of potential noisy views. How does the adaptive prior perform in such scenarios?
>
> Thank you for your valuable questions. To demonstrate the robustness of FAML,  we conducted additional experiments by introducing Gaussian noise to the data. Specifically, we applied Gaussian Noise to 20% of the views, varying the noise standard deviation from 0.5 to 1.0. The comparison results are shown below.
>
> | Method | HandWritten | Animal | Scene15 | YaleB | Caltech-101 |
> | ------ | ----------- | ------ | ------- | ----- | ----------- |
> | TLC    | 82.55       | 51.96  | 27.78   | 70.86 | 70.57       |
> | I-EDL   | 83.35       | 53.54  | 20.03   | 74.57 | 60.86       |
> | R-EDL   | 84.20       | 55.71  | 44.44   | 70.86 | 70.43       |
> | TMC    | 84.00       | 63.94  | 31.71   | 71.43 | 71.14       |
> | CCML   | 80.65       | 63.14  | 29.27   | 78.57 | 74.43       |
> | ECML   | 78.25       | 65.54  | 26.32   | 74.29 | 73.29       |
> | Ours   | 91.75       | 74.63  | 46.98   | 85.5  | 80.71       |
>
> From the results, we observe that all comparison methods fall short of maintaining robust performance during testing. In contrast, FAML achieves more robust performance across all evaluated datasets, highlighting its ability to mitigate the influence of noisy views through the adaptive calibration of the class-level Dirichlet prior. We have incorporated these results and discussions into the revised manuscript.
>
> > [Q6] Check for all possible typos in the manuscript. e.g., "bias exhibit view-specific pattern" should be "bias exhibits view-specific pattern" in Line 20.
>
> Thank you for this comment. We have fixed the typos you pointed out, and we will carefully proofread the entire manuscript to further improve its presentation in the revised manuscript.
>
> ----
>
> We hope our response could address your concerns, and we also welcome any further feedback you might provide to enhance our manuscript.

---

### Author Response · Authors · 2025-12-03
**General Response**

Dear reviewers and AC,

We sincerely appreciate your valuable time and effort spent reviewing our manuscript. We are gratified that all four reviewers highly praised the well-motivated problem, writing quality, rigorous theoretical analysis, and comprehensive experimental design of our paper. Specifically, we appreciate that the reviewers recognized:

- **Novel & Well-Motivated Problem**: Reviewers praised our identification of "Biased Evidential Multi-view Learning"  a previously **"neglected yet prevalent"** challenge (RwdE), noting it as a **"novel and well-motivated"** problem (yDqP, RwdE, bZ2G), that reflects a deep understanding of **"practical pain points"** in the field and clarifying **"key optimization directions"** for future research (Tpet).
- **Enlightening Method:** Reviewers highlighted that our method is **"enlightening"** (RwdE) and noted that the proposed components form a closed-loop optimization with **"strong targeting and novel design"** (Tpet), providing **"an effective way"** to address the practical challenges of biased evidential learning. (yDqP).

- **Strong Theoretical Foundation:** Reviewers emphasized the **"solid and complete theoretical analysis"** (Tpet, RwdE), noting that the adaptive prior is supported by **“strong theoretical grounding”** (bZ2G) and is clearly connected to the margin theory, thus offering **"an interpretable explanation"** of our adaptive prior (yDqP).

### **What We Did During Rebuttal**

During the rebuttal period, we provided responses that were thorough and sufficiently detailed to all review comments. The main points are summarized as follows:

1. **Added experiments:**
   - We added experiments on noisy multi-view datasets compared to advanced methods to show the robustness of FAML under complex scenarios (Response to the Review yDqP, Tpet).
   - We added experiments with a new evaluation metric to demonstrate the effectiveness of our method, enhancing the support to our fairness claims (Response to the Review yDqP).
   - We added experiments to illustrate how the Fairness Degree loss changes during training, providing an intuitive visualization of its underlying mechanism (Response to the Review Tpet).
   - We added experiments on common multi-view datasets under different fusion rules, showing that the adaptive prior can be acting as a plug-and-play strategy (Response to the Review RwdE).
2. **Provided clarifications:**
   - We added an explanation of the intuition of the adaptive prior to show its compensatory mechanism more explicitly (Response to the Review bZ2G).
   - We added an explanation of the motivation behind the dissonance degree, clarifying why it is appropriate for measuring discrepancies among views (Response to the Review bZ2G).
   - We added an explanations for Figure 1 to enhance the clarifications of its motivation (Response to the Review yDqP).
3. **Supplement content:**
   - We added an additional subsection on “Fairness in Machine Learning” to the related work section to strengthen the contribution and impact of our work (Response to the Review bZ2G).
   - We added more comprehensive discussion of recent EDL-based multi-view classification methods to provide clearer positioning of our work (Response to the Review yDqP).
   - We added the missing implement details regarding to the formulation of fairness loss, schedule of the balancing coefficient $\mu$ (Response to the Review Tpet, bz2G).
   - We added additional subjective logic derivations to confirm that belief and uncertainty formulations remain valid under the adaptive prior (Response to the Review RwdE).

**In summary, our rebuttal addresses every concern raised by the four reviewers with concrete experiments, detailed mechanistic analysis.** In the revised manuscript, these updates are temporarily highlighted in blue for your convenience to check. We sincerely believe that these updates may help us better deliver the benefits of the proposed FAML to the ICLR community.


----

Additionally, the authors appreciate the additional contributions of the SACs and ACs to the coordination of the discussion and assessment.

Sincerely,

The Authors of Submission 7484.

---

### Meta-Review · Area_Chair_bQcx · 2026-01-05

**Summary:**

The paper addresses a novel problem identified as "Biased Evidential Multi-view Learning" (BEML), where existing methods tend to disproportionately allocate evidence to data-rich classes, resulting in unreliable uncertainty estimates for minority classes. To mitigate this, the authors propose Fairness-Aware Multi-view Evidential Learning (FAML), utilizing a training-trajectory-based adaptive prior, a fairness constraint based on evidence variance, and an opinion alignment mechanism. The reviewers were unanimous in their positive assessment of the work, highlighting the clear motivation, the novelty of addressing evidential bias in multi-view settings, and the strong theoretical grounding (specifically the connection to margin theory). The experimental results on six real-world datasets were found to be comprehensive. Given the reviewers' consensus on the novelty of the problem and the effectiveness of the proposed solution, coupled with a robust rebuttal that addressed missing ablations and definitions, I recommend acceptance.

**Reviewer Concerns:**

The authors provided a comprehensive rebuttal that addressed the vast majority of reviewer concerns.
Addressed Concerns:
- Reviewers yDqP, Tpet: Both reviewers questioned the model's performance under noisy conditions. The authors provided new experimental results with Gaussian noise applied to views, demonstrating that FAML maintains robustness compared to baselines.
- Reviewer yDqP: The reviewer requested explicit metrics to support fairness claims. The authors introduced and reported results using a "Fairness Degree" metric, quantitatively validating their claims.
- Reviewers RwdE, bZ2G: Reviewers asked for clarification regarding the definition of fairness (class imbalance vs. sensitive attributes) and requested better positioning against fairness/subpopulation shift literature (e.g., GroupDRO, UMIX). The authors clarified the focus on label imbalance and expanded the related work section significantly.
- Reviewers RwdE, Tpet: Concerns regarding the framework's applicability to other fusion methods and performance with varying view counts were addressed through additional ablation studies showing FAML works as a plug-and-play strategy and remains effective with fewer views.
- Reviewers yDqP, bZ2G, RwdE: Various questions regarding the preservation of Subjective Logic assumptions under adaptive priors, the explicit formulation of the fairness loss ($L_{fc}$), and the schedule for the balancing coefficient ($\mu$) were answered with detailed mathematical derivations and implementation specifications.
- Reviewer yDqP: The legibility and clarity of Figure 1 were improved as requested.

Outstanding Concerns:
- There are no significant outstanding concerns. The reviewers did not post follow-up questions indicating dissatisfaction, and the authors appear to have systematically addressed all questions raised in the initial reviews.

**Reviewer Scores:**

- Reviewer yDqP (Score: 6): I believe this reviewer would likely raise their score to a 7 or 8. Their primary critiques concerned the lack of quantitative fairness metrics, insufficient literature review, and lack of noise robustness tests. The authors provided new experiments and text covering all three areas comprehensively.
- Reviewer RwdE (Score: 6): I believe this reviewer would raise their score to a 7. They explicitly stated, "If the authors can address my questions, I am willing to increase my score." The authors successfully addressed the questions regarding generalizability to other fusion methods and provided the missing implementation details for ECE calculation and hyper-parameters.
- Reviewer bZ2G (Score: 6): I believe this reviewer would raise their score to a 7. The review described the paper as "excellent" but requested "minor revisions" regarding related work expansion and intuition clarity for the adaptive prior. The authors integrated these revisions effectively.
- Reviewer Tpet (Score: 8): I believe this reviewer would maintain their score of 8. They were already very positive about the contribution ("excellent"). The additional experiments regarding noise robustness and view count analysis provided in the rebuttal would reinforce their high confidence in the acceptance of the paper.

---

### Decision · Program_Chairs · 2026-01-26

Accept (Poster)